# FeatHawkes: Scalable Feature-Based Attribution for Temporal Event Data

## Abstract

Attribution, the problem of assigning proportional responsibility for an outcome to each event in a temporal sequence of causes, is central to diverse applications ranging from marketing and seismology to sports analytics. While incorporating exogenous features substantially enhances the expressiveness of attribution models, existing approaches lack the scalability required to integrate modern machine learning. We introduce FeatHawkes, a feature-augmented Hawkes process framework for event-level attribution in continuous time. Our core contribution is a novel first-order optimization routine for Hawkes processes that leverages stochastic gradient methods, scaling favorably with both dataset size and feature dimensionality. This gradient-based formulation enables compatibility with automatic differentiation and end-to-end ML pipelines. We release FeatHawkes as an open-source Python library, and demonstrate its effectiveness through synthetic experiments and a case study on professional football data, where the framework supports what-if analyses such as quantifying the impact of replacing players in a lineup.

## 1 Introduction

Given a series of interventions, such as ads displayed to a user or treatments given to a patient, how can one determine which ones were responsible, and to what extent, for an outcome, such as a user purchase or a patient recovery? This fundamental task, known formally as *attribution*, arises naturally across many diverse fields of science and industry, such as marketing (Gaur & Bharti, 2020; Bompaire et al., 2024), neuroscience (which neurons led to activity of a given neuron) (Truccolo et al., 2005; Pillow et al., 2008), seismology (which earthquakes are fore– or aftershocks) (Ogata, 1988; Zhuang et al., 2002), but also social networks (Zhou et al., 2013), epidemiology and medicine (Rizoiu et al., 2018; Hernán & Robins, 2006), etc.

Most of these applications focus on *factual* attribution, that is, who *was* actually responsible for the outcome? In contrast, relatively little attention has been given to the *predictive* side of attribution: *what if* a different intervention were in place, how *would* the outcome have changed?

The example of sports analytics offers a compelling illustration of this distinction, which we will use throughout the paper. Assessing the *factual* contribution of an individual player to the team's success (matches won) is a crucial task for managers and analysts and has been studied in works such as Narayanan et al. (2023) or Baouan et al. (2023). Obviously, fair attribution of team success has substantial financial implications for players and is vital for managerial decision-making. Yet to optimise team composition, managers must go further: they need to evaluate how the team *would* have performed with different players. This requires a hypothetical, intervention-based understanding of individual contributions.

Predictive attribution, by necessity, builds upon a foundation of factual attribution. The simplest factual approach, *last-touch* attribution, assigns all credit to the final intervention before the outcome, i.e. to who scored the goal. While computationally simple, it clearly fails to provide a fair assessment of the team's overall contribution. In complex systems, effective attribution must be multi-touch (Shao & Li, 2011), distributing credit more equitably across all contributing interventions.

There are two approaches to addressing this problem: *physical* modelling, which, while valuable in scientific contexts, requires ad-hoc expert knowledge and complex experimentation, and *statistical* modelling, which relies solely on observed data. In statistical attribution, estimates of the conditional

probability of the outcome given a set of interventions are used to fairly distribute the *value* or *credit* created by the outcome among the interventions. These probabilities are estimated using point-process models, foregoing strong (physical) structural assumptions. Credit distribution can then be done in various ways, e.g. using Shapley values (Zhao et al., 2018) or individual treatment effects (Diemert et al., 2021).

Statistical attribution's use of estimation and fractional credit assignment focuses on being accurate on average over many events, rather than on a causal single-event basis. Naturally, then, this approach has prevailed in applications with large scale data streams, such as marketing (Shao & Li, 2011), neuroscience (Paninski, 2004; Pillow et al., 2008), seismology (Ogata, 1988; Zhuang et al., 2002), epidemiology (Rizoiu et al., 2018), and social networks (Zhou et al., 2013), etc. Nowadays, the advent of large-scale data collection and processing has opened the door to incorporating exogenous features into attribution models, making such intervention-based evaluations possible. The main driver of this movement has been online advertising, see e.g. Diemert et al. (2021).

This paper introduces a framework for predictive attribution using exogenous (feature) information. In the context of a sports team, this corresponds to data on individual players and the matches they played. The framework we propose is statistical and based on a Hawkes process structure for the event point-process. The self-excitation property of these models allows for parametric estimation of both direct (last-touch) and indirect (multi-touch) contributions, thereby overcoming the limitations of conditional expectation estimators and the combinatorial cost of Shapley values. We extend this foundation by integrating a machine learning module directly into a GPU-accelerated, gradient-based fitting routine, enabling feature-conditioned attribution through scalable end-to-end optimisation. Our open-source implementation, `FeatHawkes`, makes these tools broadly accessible. We validate the framework on synthetic benchmarks and real football data, illustrating how evaluating alternative player interventions provides actionable insights into the potential impact of transfers.

The organisation of the paper is as follows. In Section 2, we formally introduce the attribution problem, present the Hawkes process attribution framework and situate our work in context. In Section 3, we describe our estimation procedure for Hawkes processes with exogenous features, tailored for attribution under hypothetical interventions. Finally, we report our experimental results in Section 4. For illustrative clarity, we will continue to use the example of a football (soccer) team throughout the paper. Nevertheless, our framework is general and applies to temporal attribution problems in diverse domains such as marketing or seismology. Indeed, our contribution is primarily methodological, with scalability and integration with modern ML pipelines as its main strengths.

## 2 PRELIMINARIES

### 2.1 SETTING AND ATTRIBUTION PROBLEM

We consider an attribution problem in which there are $d \in \mathbb{N}$ *types* of interventions and, for simplicity, a single outcome type. We refer to these types as *dimensions* and arrange them into a vector of length $d + 1$. All of these dimensions are observed through a collection $(N^{(k)})_{k=1}^K$, $K \in \mathbb{N}$, of $(d + 1)$-dimensional counting processes[1] on $[0, +\infty)$.

In our football example, as in Baouan et al. (2023), each intervention type corresponds to a player (i.e. $d = 11$), and the associated dimension of $N^{(k)}$ counts the number of touches of the ball by this player. Thus, $N_i^{(k)}(t)$ counts the number of times player $i$ touched the ball up to time $t \geq 0$ in match $k$. Outcomes are quantities of interest for managers that can be readily computed from match data, such as the number of goals scored, or other metrics, see e.g. Baouan et al. (2023).

Given this observed data, we seek to estimate the influence exerted by dimensions $i \leq d$ on the outcome dimension $d+1$ by fitting a *Hawkes* process to the data. Let us first introduce these processes and explain how they quantify the influence between dimensions. Thereafter, in Section 3.2, we will introduce our version of this model with exogenous features.

---

[1]Meaning $N_i^{(k)}$ is almost surely a non-decreasing integer-valued function on $[0, +\infty)$, for each $i \in [d + 1]$.

## 2.2 HAWKES PROCESSES AND ATTRIBUTION

To lighten notation, let us temporarily fix $K = 1$ and drop the associated superscript $k$. We will consider that $N$ is a (random) point process on some probability space[2] and let $(\mathcal{F}_t)_{t \geq 0}$ be its filtration. The instantaneous risk of an event between time $t$ and time $t + \mathrm{d}t$ is captured by the *intensity*

$$\lambda : t \in [0, +\infty) \mapsto \lim_{h \to 0} \mathbb{E}\big[h^{-1}(N(t+h) - N(t))|\mathcal{F}_t\big] \in [0, +\infty)^{d+1} \tag{1}$$

of the process. Note that $\lambda$ is a vector-valued random function (because of conditional expectation) on $[0, +\infty)$. The counting process $N = (N_1, \ldots, N_{d+1})$, whose arrival times are denoted $(\tau_\ell^{(i)})_{\ell \in \mathbb{N}}$ for $i \in [d+1]$, is a *Hawkes process* (Hawkes, 1971) if $\lambda = (\lambda_1, \ldots, \lambda_{d+1})^\top$ is of the form

$$\lambda_i : t \in [0, +\infty) \mapsto \mu_i + \sum_{j=1}^{d+1} \sum_{\ell \in \mathbb{N}} \varphi_{i,j}(t - \tau_\ell^{(j)}) \mathbb{1}_{\{\tau_\ell^{(j)} < t\}} \tag{2}$$

for $i \in [d+1]$ and parameters $\mu \in [0, +\infty)^{d+1}$ and $\varphi_{i,j} : \mathbb{R} \to [0, +\infty)$.

This particular structure of the intensity splits events of $N$ of type $i$ into two categories: exogenous events, which arrive at a constant rate $\mu_i$ independently of the past (thus forming a Poisson process), and endogenous events, which are triggered by the process exciting itself, within the same dimension (if $\varphi_{i,i} > 0$), or from dimension $j$ to $i$ (if $\varphi_{i,j} > 0$). This self-exciting property is the key feature of Hawkes processes which will allow us to capture the influence of the interventions on the outcome.

The influence of dimensions of a Hawkes process on each other is encoded by the *branching matrix*

$$B := \left[ \int_0^\infty \varphi_{i,j}(s)\mathrm{d}s \right]_{(i,j) \in [d+1]^2},$$

which captures the magnitude of the direct excitation effects of dimensions on each other. Attributing the outcomes of dimension $d+1$ to the dimensions $i \leq d$ in proportion to the coefficients $B_{d+1,i}$ corresponds to a form of average last-touch attribution as, in effect, $B_{i,j}$ gives the average number of events of type $i$ that are triggered by a single event of type $j$.

One of the benefits of using Hawkes process models for the point process is that multi-touch attribution can be performed by computing the descendent matrix

$$D := B(I - B)^{-1},$$

which is well defined when the process is *sub-critical*, i.e. when $\int_0^{+\infty} \varphi_{i,j}(s)\mathrm{d}s < +\infty$ for all $i, j \in [d+1]$ and the spectral radius of $B$ is strictly less than 1. In effect, $D_{i,j}$ is the average total number of events of type $i$ that are triggered by a single event of type $j$. Our attribution methodology is to attribute the outcome of dimension $d+1$ to dimensions $i \leq d$ in proportion to the coefficients $D_{d+1,i}$. When extending to hypothetical or predictive attribution, we incorporate exogenous features into the process by allowing $\varphi_{i,j}$ to depend on features and parameters of each dimension pair.

## 2.3 RELATED WORK AND CHALLENGES

**Multi-touch attribution.** Most modern work on attribution focuses on multi-touch attribution (MTA) in marketing and advertising. MTA represents a methodological advance over simple heuristics like "last-click" or "first-click" by distributing credit across multiple touchpoints in a customer's journey (Li & Kannan, 2014; Xu et al., 2014). Early heuristic models (e.g., linear, time-decayed, or U/W-shaped rules) are valued for their simplicity but criticised for lacking empirical grounding (Shao & Li, 2011; Berman, 2018; El Mekkaoui & Benyoussef, 2024; Mrad & Hnich, 2024).

To overcome these limitations, data-driven models were introduced. Probabilistic models, such as those based on Markov chains, infer the relative importance of each interaction from user path data (Anderl et al., 2016; Dalessandro et al., 2015). Causal methods, have also been proposed, on the basis of the do-calculus (Athey & Imbens, 2017; Bottou et al., 2013). Finally, game-theoretic methods based on Shapley values allocate credit by computing marginal contributions across possible scenarios (Shao & Li, 2011; Datta et al., 2017). More recently, machine learning approaches have been used to capture complex, non-linear interactions between channels at scale (Zhang et al., 2014).

---

[2]We defer a more formal series of definitions and measurability arguments to Appendix A.

**Hawkes processes.** Since their introduction by Hawkes (1971), Hawkes processes have become a cornerstone in both theoretical and applied modelling of self-exciting phenomena. Classical applications include demographic modelling via Galton-Watson processes (Neves & Moreira, 2006) and earthquake aftershock sequences (Ogata, 1988; Iwata, 2025; Kwon et al., 2023). More recently, they have been applied in neuroscience (Truccolo et al., 2005; Pillow et al., 2008), social networks (Zhou et al., 2013; Rizoiu et al., 2018), and epidemiology (Rizoiu et al., 2018), among others.

Theoretical work has primarily focused on the probabilistic structure of Hawkes processes (Hawkes, 1971), with significant contributions to their statistical estimation via maximum likelihood (Ogata, 1978; Ozaki, 1979), EM algorithms (Veen & Schoenberg, 2008; Lewis & Mohler, 2011), and Bayesian approaches (Rasmussen, 2013; Linderman & Adams, 2014; Donnet et al., 2020). Hawkes models have also seen widespread adoption in finance, where they are used to capture high-frequency trading dynamics and market microstructure (Jaisson & Rosenbaum, 2015; 2016; Dandapani et al., 2021; Rosenbaum & Tomas, 2021); see also the survey by Bacry et al. (2015).

Beyond the classical setting, extensions to multidimensional and marked Hawkes processes are well-established (Laub et al., 2021), and recent developments include nonparametric inference and structural variants (Bonnet et al., 2023). Exogenous features have also been incorporated into Hawkes process, either in the form of spatio-temporal models or Neural Hawkes processes. The former encodes both spatial (feature-space) and temporal evolution (Bernabeu et al., 2025). The latter integrate recurrent neural architectures into the excitation probabilities of the Hawkes process (Mei & Eisner, 2017; Zhang et al., 2020; Shchur et al., 2021). The expressivity of these models allows them to capture complex non-linear endogenous dynamics. Despite their integration of features, these models are not attribution models. Our contribution is therefore complementary: rather than maximising expressiveness of dynamics, we focus on attribution with exogenous features and scalable inference.

## 2.4 Challenges and Contributions

**Challenges.** Despite significant progress, two fundamental challenges remain unresolved in the literature on both attribution and Hawkes processes. First, statistical attribution models are inherently limited in their ability to incorporate exogenous features, which are essential for evaluating alternative interventions, a core requirement for predictive attribution. Without the ability to model how changes in external covariates (e.g. player traits, treatment, or marketing interventions) influence outcomes, these models cannot support the evaluation or optimisation of interventions. Second, the numerical methods traditionally used to fit Hawkes models are inefficient and outdated, rendering them unsuitable for large-scale applications. In particular, they are ill-equipped to integrate with machine learning pipelines, where efficient, differentiable and scalable training procedures are critical.

**Contributions.** This paper addresses both challenges directly. First, we introduce a feature-augmented Hawkes model in which the excitation kernel is governed by feature-dependent (random) coefficients. This formulation enables machine learning models to learn complex attribution rules from exogenous information, facilitating predictive evaluation of interventions. Second, we develop a stochastic gradient ascent procedure for penalised conditional likelihood maximization. Unlike EM or MLE approaches, our method is GPU-compatible, scalable, and differentiable, making it suitable for integration with modern ML workflows. These innovations are packaged in our open-source Python library `FeatHawkes`, designed for accessibility and real-world deployment. We validate our approach both on synthetic benchmarks and real football data, demonstrating both computational efficiency and the practical utility of predictive attribution in complex decision-making contexts.

## 3 Statistical Estimation and Model Fitting

### 3.1 Maximum Likelihood Estimation for Hawkes Processes

Statistical attribution is an unsupervised learning problem, since by definition the true cause of each event is unobserved. The use of a Hawkes process to model the event times allows learning of relationships between interventions (or features) and outcomes using its likelihood model as a loss function. We present this loss function and discuss its estimation in this section.

In order to make optimisation efficient, we will focus on a parametric Hawkes process, and in particular on exponential kernels which imposes $\varphi_{i,j} : t \mapsto \alpha_{i,j}\beta_{i,j}e^{-\beta_{i,j}t}$ for every $(i,j) \in [d+1]^2$, and for some parameters $(\alpha_{i,j}, \beta_{i,j}) \in [0, +\infty)^2$. Exponential kernels are the kernel of choice in most applications, see e.g. Laub et al. (2021, § 3.4), as they are simple to fit via maximum likelihood (see below) and to interpret. Indeed, $\alpha_{i,j}$ gives the magnitude of the excitation effect from $j$ to $i$ while $\beta_{i,j}$ gives the rate of decay of this effect. Practical examples also show that exponential decay is well-suited to many applications (Ogata, 1988). This model has straightforward attribution as $B = \alpha$.

Given a single trajectory, up to a time $T \in [0, +\infty)$, of an exponential Hawkes process $N$ with intensity $\lambda$, the negative log-likelihood of the parameters $(\mu, \alpha, \beta) := [\mu_i, \alpha_{i,j}, \beta_{i,j}]_{i,j}$ is given by

$$\mathcal{L}(N; \mu, \alpha, \beta) = -\sum_{i=1}^{d+1} \sum_{\ell=1}^{N_i(T)} \log \lambda_i \left( \tau_\ell^{(i)} \right) - \int_0^T \lambda_i(t)\mathrm{d}t \,. \tag{3}$$

Inspection of (2) in the context of (3) highlights one of the clear benefits of an exponential kernel: the time integral of the intensity, a quantity known as the *compensator* can be computed in closed form without the need for numerical integration, see Ogata (1978) or Daley & Vere-Jones (2007, Ch. 7).

Minimisation of $\mathcal{L}(N; \cdot)$ to fit a Hawkes process to the data of $N$ by maximum likelihood is complicated by two factors: (i) the non-convexity of the $\mathcal{L}(N; \cdot)$; (ii) the need to respect positivity constraints on the parameters. The first issue is unfortunately unavoidable and makes implementation quite sensitive to initialisation. Nevertheless, $\mathcal{L}(N; \cdot)$ is a smooth function of the parameters on the interior of the parameter space, which is sufficient to use gradient-based optimisation methods.

Unfortunately, the use of (simple) gradient-based methods is not appropriate for constrained minimisation problems. To avoid this issue, we propose the introduction of a barrier function

$$\Upsilon : (\mu, \alpha, \beta) \mapsto -\sum_{i=1}^{d+1} \left( \log(\mu_i) + \sum_{j=1}^{d+1} \log(\alpha_{i,j}) + \log(\beta_{i,j}) \right) \,,$$

using the convention $\log(\cdot) = -\infty$ on $(-\infty, 0]$.

In summary, our method for maximum likelihood estimation of Hawkes processes with exponential kernels solves the following optimisation problem:

$$\min_{\mu, \alpha, \beta} \mathcal{L}(N; \mu, \alpha, \beta) + \eta \Upsilon(\mu, \alpha, \beta) \,, \tag{4}$$

for $\eta > 0$. Minimising in (4) requires gradient-based optimisation, but is bottlenecked by the nested sums in the gradient of (3). The speed of any implementation hinges on how efficiently these sums are computed. Our `FeatHawkes` library leverages GPU acceleration via `PyTorch` to overcome the performance limits of traditional CPU-based methods, and includes optional $L_1$ penalisation of the likelihood (see Appendix C.1 for details).

Beyond implementation tricks, the key to fast and scalable fitting lies in the choice of optimisation method. Full gradient (or Hessian) computation becomes prohibitive for multiple or long trajectories, making stochastic or partial-gradient methods more efficient. `FeatHawkes` adopts an adaptive mini-batch gradient strategy using the `PyTorch` optimiser framework, avoiding the costly operations used in earlier algorithms like Ogata (1978) and Veen & Schoenberg (2008) (see Appendix C.2).

## 3.2 Incorporating Features and Fitting Machine Learning Models to Data

Let us return to our general setting, consisting of a dataset of trajectories $N^{(k)}$, $k = 1, \ldots, K$, each with length $T_k > 0$. We assume that each trajectory has a fixed, time-invariant[3], collection of features $(Z^{(k)}, (X_i^{(k)})_{i \in [d+1]})$ associated with the whole trajectory and individual dimensions (respectively). We consider that theses features affect the intensity of the Hawkes process $N^{(k)}$ via (1) with kernel $\varphi_{i,j} := \varphi_{\theta_{i,j}}(X_i^{(k)}, X_j^{(k)}, Z^{(k)})$ for a fixed $\varphi$ and $\theta_{i,j}$ parameters of the pair $(i,j)$ to be learnt. Any parametric ML model can be used here for $\varphi$, provided it outputs probabilities, i.e. $\varphi : \mathbb{R} \to [0, 1)$.

---

[3]If features are time-dependent, the time-integral of $\lambda$ in (3) may lose its closed form, dramatically increasing numerical complexity.

For the sake of exposition, we will focus on a logistic regression model in which only the $\alpha$ coefficients are dependent on features and $\beta_{i,j} = \beta_i \in (0, +\infty)$ for every $(i,j) \in [d+1]^2$. The method extends naturally if $\mu$ or $\beta$ are also dependent on features, at the cost of increased complexity in the fitting procedure. Specifically, we will write

$$\alpha_{i,j} = \alpha_{i,j}(\gamma, \theta; X^{(k)}, Z^{(k)}) := \sigma\left(\gamma_{i,j} + {\theta_{i,j}^{(0)}}^\top Z^{(k)} + {\theta_{i,j}^{(1)}}^\top X_i^{(k)} + {\theta_{i,j}^{(2)}}^\top X_j^{(k)}\right), \quad (5)$$

for some coefficients $\gamma_{i,j} \in \mathbb{R}$, $\theta_{i,j}^{(0)} \in \mathbb{R}^{d_z}$, $\theta_{i,j}^{(1)} \in \mathbb{R}^{d_i}$, and $\theta_{i,j}^{(2)} \in \mathbb{R}^{d_j}$, $(i,j) \in [d+1]^2$, with $\theta_{i,j} := (\theta_{i,j}^{(0)}, \theta_{i,j}^{(1)}, \theta_{i,j}^{(2)})$ and $\sigma : \mathbb{R} \to (0,1)$ denoting the logistic function. This statistical model is identifiable (see Proposition A.1 in Appendix A), up to convex combinations of the values of $\theta_{i,i}^{(1)}$ and $\theta_{i,i}^{(2)}$, an issue circumvented in practice by imposing $\theta_{i,i}^{(1)} = \theta_{i,i}^{(2)}$ without loss of generality.

Combining (4) and (5) allows us to write a (conditional) likelihood function

$$\mathcal{L}(N; \mu, \gamma, \theta, \beta | X, Z) := \sum_{k=1}^{K} \mathcal{L}(N^{(k)}; \mu, \alpha(\gamma, \theta; X^{(k)}, Z^{(k)}), \beta) \quad (6)$$

for the parameters of the logistic regression and the Hawkes process in an end-to-end procedure. Since $\alpha$ is in $(0,1)$ we may remove it from $\Upsilon$, yielding the optimisation problem

$$\min_{\mu, \theta, \beta} \mathcal{L}(N; \mu, \theta, \beta | X, Z) - \eta \sum_{i=1}^{d+1} \log(\mu_i) + \sum_{j=1}^{d+1} \log(\beta_{i,j}).$$

Note that alternative optimisation methods such as Frank–Wolfe (Zhao, 2025), dual coordinate ascent (Bompaire et al., 2018), or projection-free adaptive gradients (Combettes et al., 2020) are effective in convex settings with fixed decay parameters ($\beta$), but are not directly applicable in our general, non-convex case where all $(d+1)^2$ coefficients must be learned jointly.

## 4 SIMULATION STUDY

### 4.1 NUMERICS OF HAWKES PROCESS FITTING

Obtaining quantitative convergence guarantees for Hawkes process fitting algorithms remains challenging due to the non-convexity of the negative log-likelihood $\mathcal{L}$, even without features. To date, no non-asymptotic convergence results are known. As a result, algorithmic comparisons must rely on empirical evaluation. To avoid the confounding effects of real-world data, we use controlled simulation studies. Fortunately, simulating Hawkes processes is well understood following the method of Ogata (1978), which we implement in `FeatHawkes` and describe in Appendix B.

Our first simulation experiment, summarised in Figure 1, consists in recovering the parameters

$$\tilde{\mu} = \begin{pmatrix} 0.05 \\ 0.1 \\ 0.15 \\ 0.2 \end{pmatrix}, \quad \tilde{\alpha} = \begin{pmatrix} 0 & 0.1 & 0.05 & 0.03 \\ 0.1 & 0 & 0.1 & 0.05 \\ 0.05 & 0.1 & 0 & 0.1 \\ 0.03 & 0.05 & 0.1 & 0 \end{pmatrix}, \quad \text{and } \tilde{\beta} = 1.4$$

from a single randomly-generated trajectory containing $\mathcal{N} \in \mathbb{N}$ events, for a range of values of $\mathcal{N}$. Because of the comparable results and computational cost of existing algorithms, we limit ourselves to the two most common methods as benchmarks: the Maximum-Likelihood algorithm of Ogata (1978), which we denote `MLE`, and the Expectation-Maximisation algorithm of Veen & Schoenberg (2008), which we denote `EM`. We report the averages of five instances of this experiment in Figure 1. Absent standard stopping heuristics, all algorithms were considered to have terminated when their respective loss criteria failed to improve by more than $10^{-5}$ between two subsequent steps.

Figure 1a shows our algorithm achieves the accuracy of `MLE` (omitted for clarity as indistinguishable) and better than `EM`. However, Figure 1b highlights stark differences in computational efficiency: `MLE` is 100 times slower than `EM`, and both scale quadratically with the time horizon. In contrast, our

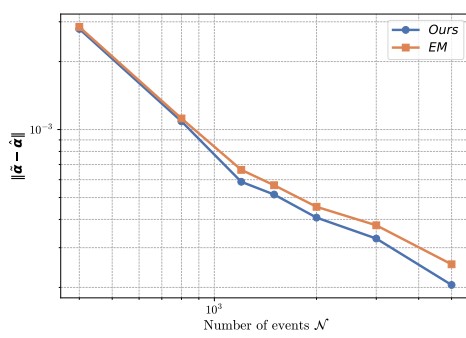
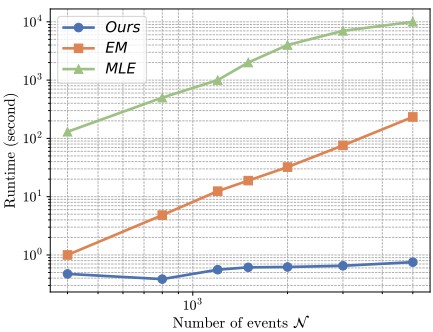

(a) Estimation error per sample size $\mathcal{N}$. (Ogata (1978) not shown as indistinguishable from ours.)

(b) Runtime as a function of the sample size $\mathcal{N}$.

Figure 1: Quantitative comparison of quality and efficiency of Hawkes process fitting algorithms.

method leverages stochastic gradients and adaptive batching (see Section 3.1), yielding a sublinear cost growth of $\mathcal{N}^{\frac{1}{2}}$ and already outperforming EM by two orders of magnitude at $\mathcal{N} = 5000$.

Realistic attribution tasks, such as modelling a football team ($d = 11$), demand scalability in the number of dimensions $d + 1$. As shown in Figure 2, EM scales poorly with $d$; MLE is excluded due to prohibitive runtime. In contrast, our algorithm scales roughly linearly and is already 100 times faster than EM for $d = 6$. When incorporating features for attribution, the baseline runtime from Figure 1b is further compounded by the cost of fitting the regression model over $(\gamma, \theta)$. As Figure 2 makes clear, only our gradient-based fitting method, which is readily compatible with machine learning models, offers practical scalability for real-world applications.

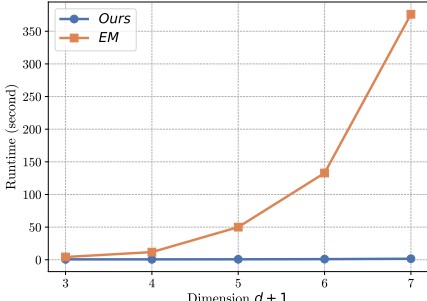
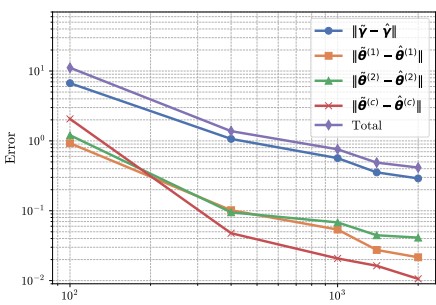

Figure 2: Runtime as a function of the dimension $d + 1$.

Figure 3: Estimation error as a function of the number of trajectories $K$.

**Fitting with exogenous features and logistic regression.** To demonstrate the convergence of our method with features, we fix $d = 2$ and coefficients

$$\tilde{\mu} := \begin{pmatrix} 0.1 \\ 0.15 \\ 0.2 \end{pmatrix}, \ \tilde{\gamma} := \begin{pmatrix} -1 & -0.5 & -1 \\ -1 & -0.5 & -1 \\ -0.5 & 0 & 0 \end{pmatrix}, \ \tilde{\theta}^{(1)} := \begin{pmatrix} 0 & 0 & 0 \\ 0 & -0.75 & -0.5 \\ 0 & -1 & 0 \end{pmatrix},$$

$$\tilde{\theta}^{(2)} := \begin{pmatrix} 0 & 0 & 0.75 \\ 0.5 & 0 & -0.5 \\ 0.75 & -0.25 & 0 \end{pmatrix}, \ \tilde{\beta} := \begin{pmatrix} 0.8 & 0.8 & 0.8 \\ 1 & 1 & 1 \\ 1.2 & 1.2 & 1.2 \end{pmatrix}. \tag{7}$$

We then simulate $\kappa \in \mathbb{N}$ copies of randomly sampled features $(\tilde{X}_i)_{i \in [d+1]}$ with each $\tilde{X}_i \in \mathbb{R}$ (ignoring $Z$ for simplicity). We then simulate 20 trajectories for each sampled feature of the resulting conditional Hawkes process, i.e. (5) with $(\tilde{\gamma}, \tilde{\theta}, 0, (\tilde{X}_i)_{i \in [d+1]})$ instead of $(\gamma, Z^{(k)}, (X_i^{(k)})_{i \in [d+1]})$ by

using the method of Ogata (1981). Figure 3 shows the resulting error[4] curves as a function of the total number of trajectories $K = 20\kappa$ used in the estimation. The overall error decreases as $K^{-\frac{1}{2}}$, indicating an efficient (parametric-rate) learning procedure. Using the same protocol, we further evaluate whether the fitted attribution matrices $(\hat{\alpha}, \hat{D})$ successfully recover the generative matrices implied by $(\tilde{\gamma}, \tilde{\theta})$ in the data-generating process. The results, presented in Figure 4, in Appendix D.1, show an error for both of order $K^{-\frac{1}{2}}$, demonstrating the robustness and data-efficiency of our method.

Finally, we compare our credit attribution method through $\hat{D}$ to alternative credit assignments from the point-process estimates $\hat{\alpha}$ using two classical attribution methods, Shapley values and uplift. We describe these experimental details in and give results in Appendix D.2 Table 4. These results show that, at least when the ground-truth generative model is a Hawkes process, our method outperforms the baselines by a factor of two in terms of the error with respect to $D$.

## 4.2 PREDICTIVE ATTRIBUTION IN FOOTBALL

Sports are often viewed by the public as purely athletic endeavours, yet, at the professional level, they have become highly data-driven disciplines. In team sports in particular, performance data has been widely adopted by managers seeking to optimise their teams. A central challenge in this endeavour is understanding how individual players contribute to collective success—an attribution problem.

To illustrate the effectiveness of our predictive attribution methodology, we consider one of the most high-profile recent transfers in European football: the move of French forward Kylian Mbappé from France's Paris Saint-Germain (PSG) to Spain's Real Madrid (RM). Using match data from the 2023–2024 and 2024–2025 domestic leagues and UEFA Champions League, we assess whether our model accurately predicted Mbappé's performance at RM, and whether his contribution exceeded expectations or not. Before that, we describe the data and validate our method's attribution.

**Data.**    To train our attribution model, we compiled a dataset comprising the 60 matches played in the second half of the 2023–2024 season by four leading European clubs with comparable characteristics (e.g. budget, composition, tactical approach): PSG, RM, Manchester City, and Liverpool. The dataset, proprietary to the football analytics firm Footovision, contains around 24000 individual events across 60 match trajectories.

We build our Hawkes process model by assigning each player to a dimension $i \in [d]$ corresponding to their role on the pitch. For the outcome dimension, we follow standard practice in football analytics (Baouan et al., 2023), identifying times when the ball enters the opponent's penalty area[5]. This proxy outcome smooths the randomness associated with goals while retaining meaningful signals of attacking value. We record an event at time $t$ in dimension $i$ when player $i$ touches the ball, and an event in the outcome dimension $d + 1$ when the ball enters the penalty box.

To avoid feature leakage, we use the first half of the 2023–2024 season exclusively to construct the features $X^{(v)}$ and $Z^{(v)}$, for each team $v \in [4]$. These features remain fixed across that team's matches in the second half of the season. They include statistics on players, teams, and leagues, all derived from proprietary Footovision data. Full details on data processing and feature selection are provided in Appendix D.4. We fit the Hawkes process and logistic regression model to the second-half matches by minimising (6), using a deterministic batched ADAM optimiser with early stopping. Further implementation details are provided in Appendix D.3. The resulting fitted coefficients for each team $v \in [4]$ are denoted by $(\breve{\mu}_v, \breve{\gamma}_v, \breve{\theta}_v, \breve{\beta}_v)$.

**Empirical validation.**    In the absence of ground truth data, to validate the quality of the predictive attribution, we will compare its predictions to a feature-less model fitted retro-actively. Namely, we estimate the excitation coefficients for each team $v \in [4]$ during the 2024–2025 season using the featureless Hawkes model from Section 3.1, yielding parameters $(\hat{\mu}_v, \hat{\alpha}_v, \hat{\beta}_v)$. We then compare to

---

[4]Because of the identifiability problem, we compare $\tilde{\theta}_i^{(c)} := \tilde{\theta}_{i,i}^{(1)} + \tilde{\theta}_{i,i}^{(2)}$ to $\hat{\theta}_i^{(c)} := \hat{\theta}_{i,i}^{(1)} + \hat{\theta}_{i,i}^{(2)}$.

[5]We also include long-range shots with an estimated scoring probability of at least 25% to mitigate sparsity.

this the predictions made using $(\breve{\mu}_v, \breve{\gamma}_v, \breve{\theta}_v, \breve{\beta}_v)$ on the 2024–2025 features through the average error

$$\mathcal{E} := \frac{1}{4} \sum_{v=1}^{4} \|\hat{\mu}_v - \breve{\mu}_v\|^2 + \left\| \hat{\alpha}_v - \sigma \left( \breve{\gamma}_{i,j} + \breve{\theta}_{i,j}^{(0)\top} Z^{(v)} + \breve{\theta}_{i,j}^{(1)\top} X_i^{(v)} + \breve{\theta}_{i,j}^{(2)\top} X_j^{(v)} \right) \right\|^2 + \left\| \hat{\beta}_v - \breve{\beta}_v \right\|^2.$$

The results, summarised in Table 1, show a small average error of $\mathcal{E} \approx 0.13$ across 600 parameters, confirming that the model can reliably infer performance from features. Importantly, the accuracy of predictions for $\mu$ and $\beta$ suggests that team performance is temporally stable enough to make such predictions meaningful. The higher errors for Manchester City and Real Madrid are attributable to their underperformance during the 2024–2025 season.

Table 1: Discrepancy between the estimated coefficients of our featureless model for the 2024–2025 season and the coefficients predicted using the full model fit on the 2023–2024 data, for four major European clubs. The average total corresponds to $\mathcal{E}$.

| Club ($v$) | Error Terms | | | |
|---|---|---|---|---|
| | $\|\hat{\mu}_v - \breve{\mu}_v\|^2$ | $\|\hat{\alpha}_v - \sigma(\cdots)\|^2$ | $\|\hat{\beta}_v - \breve{\beta}_v\|^2$ | **Total** |
| PSG | 0.0102 | 0.095 | 0.0160 | 0.1212 |
| Manchester City | 0.0096 | 0.130 | 0.0090 | 0.1486 |
| Liverpool | 0.0108 | 0.088 | 0.0120 | 0.1108 |
| Real Madrid | 0.0099 | 0.120 | 0.0140 | 0.1439 |
| **Average** | **0.0101** | **0.1083** | **0.0128** | **0.1311** |

**Predictive attribution**  Our model estimates Kylian Mbappé's (#9) direct excitation of scoring opportunities during the 2023–2024 season with PSG to be $0.13$ (see Figure 9), with a total influence score of $0.30$ (see Figure 10). For his transfer to Real Madrid, the model predicted using the RM feature a low excitation $\hat{\alpha}_{12,9} = 0.05$ (see Figure 13) and reduced total influence of $0.17$ (see Figure 14). While the overall importance was well forecasted (see Figure 12), the direct excitation was underestimated as Mbappé ultimately posted a factual excitation of $0.10$ at RM (see Figure 11). This gap is consistent with post-transfer behavioural shifts.

Our framework also supports hypothetical roster changes. Had Mbappé joined Liverpool to replace Cody Gakpo (#8), his predicted direct contribution ($0.09$) would have been similar to that with PSG; however, the overall impact would have been lower due to diminished excitation of teammates—a difference tied to role-specific features rather than raw quality (see Appendix D.5).

## 5 CONCLUSION

We have introduced a novel attribution methodology grounded in Hawkes processes, enabling feature-based reasoning in complex event-driven systems. Our framework addresses the central challenge of extrapolating attribution beyond observed data, crucial for informing policy and strategic interventions in high-dimensional interactive settings such as team sports. In addition, we developed a performant, scalable, and GPU-accelerated software library integrated with `PyTorch`, and validated our approach empirically on both synthetic and real-world football data.

Looking ahead, this work provides a foundation for developing more advanced models of interpretable, data-driven attribution in complex systems, with broad applicability across domains. For example, our current approach captures only first-order effects and does not account for higher-order interactions such as how variations in one feature may alter the dependencies of interactions among other features. Extending the framework to learn the excitation structure $\alpha$ from all features jointly would be a worthwhile direction of future work to model these effects. Likewise, although we use a single outcome variable for offensive threat, the framework naturally extends to multiple outcomes, and discovering relevant outcome combinations for applications presents a fruitful direction.

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

## TECHNICAL APPENDICES AND SUPPLEMENTARY MATERIAL

These appendices are independent of each other and organised in the following manner. Section A below completes the preliminaries by providing rigourous definitions of Hawkes processes and the surrounding probabilistic framework. Sections B and C give mathematical and implementation details of our framework and `FeatHawkes`, respectively focusing on the simulation of Hawkes processes (used for our synthetic experiments) and on maximum likelihood fitting (accompanying the statistics section) respectively. Finally, Section D contains extensive details on the experiments, both synthetic and real-data.

## A  DEFINITIONS AND NOTATIONS

### A.1  PROBABILISTIC FRAMEWORK

We first consider a probability space $(\Omega_X, \mathcal{F}_X, \mathbb{P}_X)$ with $\Omega_X := \mathbb{R}^{d_z + \sum_{i=1}^{d+1} d_i}$, endowed with a random variable $W := (Z, (X_i)_{i \in [d+1]})$, with $Z$ being $\mathbb{R}^{d_z}$-valued and each $X_i$, $i \in [d+1]$, being $\mathbb{R}^{d_i}$-valued. For the sake of completeness, we take $\mathcal{F}_X$ to be the Borel $\sigma$-algebra on $\Omega_X$.

In parallel, we consider the Skorokhod space $\mathbb{D}([0, +\infty), \mathbb{R}^{d+1})$ of all càdlàg (right-continuous with left limits) $(d+1)$-dimensional real vector-valued functions on $[0, +\infty)$. We let $\Omega_H := \mathbb{D}([0, +\infty), \mathbb{R}^{d+1})$ and $\mathcal{F}_H$ be the completion of its Borel $\sigma$-algebra.

Let $\Theta \subset \mathbb{R}^{d_\vartheta}$ be a parameter set and $w \in \Omega_X$, $w := (z, x)$ and let $\vartheta := (\mu, \theta, \beta)$, for $\mu \in [0, +\infty)^{d+1}$, $\beta \in (0, +\infty)^{d+1}$, and $\theta := (\gamma, \theta^{(0)}, \theta^{(1)}, \theta^{(2)}) \in \mathbb{R}^{(d+1)^2} \times (\mathbb{R}^{d_z})^{d+1} \times \prod_j \prod_i \mathbb{R}^{d_i} \times \prod_j \prod_i \mathbb{R}^{d_i}$. Let $N^{(\vartheta, w)}$ be a random variable which forms a counting process on $(\Omega_H, \mathcal{F}_H)$ and define on $[0, +\infty)$, for any locally bounded measurable function $f : [0, +\infty) \to \mathbb{R}$, the stochastic integral

$$\int_0^\cdot f(s) \mathrm{d}N^{(\vartheta, w)}(s) := \sum_{s < \cdot} f(s) \mathbb{1}_{\{N^{(\vartheta, w)}(s-) \neq N^{(\vartheta, w)}(s)\}},$$

wherein $N^{(\vartheta, w)}(s-) := \lim_{\epsilon \downarrow 0} N^{(\vartheta, w)}(s - \epsilon)$ is the left limit of $N^{(\vartheta, w)}$ at $s$.

Under any probability measure $\tilde{\mathbb{P}}$ on $(\Omega_H, \mathcal{F}_H)$, the $\tilde{\mathbb{P}}$-compensator of $N^{(\vartheta, w)}$ is the unique predictable process $\Lambda^{(\vartheta, w)}$ with $\Lambda^{(\vartheta, w)}(0) \stackrel{\text{a.s.}}{=} 0$ such that $N^{(\vartheta, w)} - \Lambda^{(\vartheta, w)}$ is a local martingale. For any $f$ locally bounded and measurable,

$$\tilde{\mathbb{E}} \left[ \int_0^\cdot f(s) \mathrm{d}N_s^{(\vartheta, w)} \right] = \tilde{\mathbb{E}} \left[ \int_0^\cdot f(s) \mathrm{d}\Lambda^{(\vartheta, w)} \right]$$

wherein $\tilde{\mathbb{E}}$ denotes the expectation with respect to $\tilde{\mathbb{P}}$.

Let $\mathbb{P}^{(\vartheta, w)}$ be a probability measure such that $N^{\vartheta, w}$ is an exponential Hawkes process parametrised by $\vartheta$. In other words, such that its compensator is of the form

$$\Lambda^{(\vartheta, w)} : t \in [0, +\infty) \mapsto \int_0^t \lambda^{(\vartheta, w)}(s) \mathrm{d}s$$

wherein $\lambda^{(\vartheta, w)} := (\lambda_1^{(\vartheta, x)}, \dots, \lambda_{d+1}^{(\vartheta, x)})^\top$ and

$$\lambda_i^{(\vartheta, w)} : t \in [0, +\infty) \mapsto \mu_i + \sum_{j=1}^{d+1} \int \varphi_{i,j}^{\vartheta, w}(s - u) \mathrm{d}N^{(\vartheta, w)}(u)$$

for $i \in [d+1]$ and the parametric collection of functions

$$\varphi_{i,j}^{\vartheta, w} : t \in [0, +\infty) \mapsto \sigma \left( \gamma_{i,j} + {\theta_{i,j}^{(0)}}^\top z + {\theta_{i,j}^{(1)}}^\top x_i + {\theta_{i,j}^{(2)}}^\top x_j \right) \beta_{i,j} e^{-\beta_{i,j} t} \in [0, +\infty),$$

for any $(i, j) \in [d+1]^2$.

Now, we move from the conditional measure family $(\mathbb{P}^{(\vartheta,w)})_{\vartheta \in \Theta}$ to the joint family

$$\mathbb{P}_\vartheta : S_1 \times S_2 \in \mathcal{F}_H \times \mathcal{F}_X \mapsto \int_{S_2} \int_{S_1} \mathrm{d}\mathbb{P}^{(\vartheta,W)} \mathrm{d}\mathbb{P}_X \,,$$

and consider the statistical model $(\Omega, \mathcal{F}, (\mathbb{P}_\vartheta)_{\vartheta \in \Theta})$.

## A.2 Model Specification and Identifiability

**Proposition A.1.** *Suppose the measure $\mathbb{P}_X$ is non-degenerate, in the sense that there are at least two disjoint $\mathbb{P}_X$-non-null subsets $(A, B)$ of $\mathcal{F}_X$ such that there is no affine function $\phi : A \to B$ with $W|_B = \phi(W|_A)$. Then, the statistical model $(\Omega, \mathcal{F}, (\mathbb{P}_\vartheta)_{\vartheta \in \bar{\Theta}})$ is identifiable. In other words, for any $(\vartheta, \tilde{\vartheta}) \in \bar{\Theta}^2$, $\mathbb{P}_\vartheta \neq \mathbb{P}_{\tilde{\vartheta}}$.*

*Proof.* Let $(\vartheta, \tilde{\vartheta}) \in \Theta^2$ and let $\mathbb{P}_\vartheta$ and $\mathbb{P}_{\tilde{\vartheta}}$ be their induced measures. Denote $\vartheta := (\mu, (\gamma, \theta^{(0)}, \theta^{(1)}, \theta^{(2)}), \beta)$ and $\tilde{\vartheta} := (\tilde{\mu}, (\tilde{\gamma}, \tilde{\theta}^{(0)}, \tilde{\theta}^{(1)}, \tilde{\theta}^{(2)}), \tilde{\beta})$. Assume for a contradiction that $\mathbb{P}_\vartheta = \mathbb{P}_{\tilde{\vartheta}}$ which implies $\mathbb{P}^{(\vartheta,\cdot)} = \mathbb{P}^{(\tilde{\vartheta},\cdot)}$ on any $\mathbb{P}_X$-non-null set of $\in \Omega_X$. As Hawkes measures, both $\mathbb{P}^{(\vartheta,w)}$ and $\mathbb{P}^{(\tilde{\vartheta},w)}$ are absolutely continuous with respect to $\mathbb{P}_w^*$, the homogenous Poisson measure of rate 1, thus

$$\frac{\mathrm{d}\mathbb{P}^{(\vartheta,w)}}{\mathrm{d}\mathbb{P}_w^*} \overset{\text{a.s.}}{=} \frac{\mathrm{d}\mathbb{P}^{(\tilde{\vartheta},w)}}{\mathrm{d}\mathbb{P}_w^*} \,.$$

Recalling the form of the likelihood from (3), for $\mathbb{P}_w^*$-almost every $N \in \Omega_H$, we obtain

$$\sum_{i=1}^{d+1} \int_0^\cdot \log \lambda_i^{(\vartheta,w)}(s)\mathrm{d}N(s) - \int_0^\cdot \lambda^{(\vartheta,w)}(s)\mathrm{d}s = \sum_{i=1}^{d+1} \int_0^\cdot \log \lambda_i^{(\tilde{\vartheta},w)}(s)\mathrm{d}N(s) - \int_0^\cdot \lambda^{(\tilde{\vartheta},w)}(s)\mathrm{d}s$$

on $[0, +\infty)$. Let $\tau = \inf\{t \in [0, +\infty) : N(t) > 0\}$ be the first jump time of $N$, which is almost surely strictly positive. Thus, almost surely, $\lambda^{(\vartheta,w)} = \lambda^{(\tilde{\vartheta},w)}$ on $[0, \tau)$, thus the likelihoods are equal on $[0, \tau]$ and by proceeding likewise between each jump,

$$\lambda^{(\vartheta,w)} \overset{a.s.}{=} \lambda^{(\tilde{\vartheta},w)} \text{ on } [0, +\infty) \,. \tag{8}$$

On $[0, \tau)$, (8) shows that $\mu = \tilde{\mu}$, while for $t > \tau$ it shows that

$$\sum_{j=1}^{d+1} \sum_{\ell \in \mathbb{N}} \alpha_{i,j}(\gamma, \theta; x, z)\beta_{i,j} e^{-\beta_{i,j}(t-\tau_\ell^{(j)})} \mathbb{1}_{\{\tau_\ell^{(j)} < t\}}$$

$$= \sum_{j=1}^{d+1} \sum_{\ell \in \mathbb{N}} \alpha_{i,j}(\tilde{\gamma}, \tilde{\theta}; x, z)\tilde{\beta}_{i,j} e^{-\tilde{\beta}_{i,j}(t-\tau_\ell^{(j)})} \mathbb{1}_{\{\tau_\ell^{(j)} < t\}}$$

where $\tau_\ell^{(j)} := \inf\{t \in [0, +\infty) : N_j(t) = \ell\}$ for $(j, \ell) \in [d+1] \times \mathbb{N}$ and $w := (z, x)$. By non-degeneracy of $\mathbb{P}_X$, $\alpha_{i,j}(\gamma, \theta; x, z) \neq \alpha_{i,j}(\tilde{\gamma}, \tilde{\theta}; x, z)$, leaving us with two countable combinations of families of functions of the form

$$t \mapsto a_\upsilon e^{-b_\upsilon(t-s)} \text{ and } t \mapsto \tilde{a}_\upsilon e^{-\tilde{b}_\upsilon(t-s)}$$

as indexed by a shared index $s \in [0, +\infty)$, for some $\upsilon \in [d+1]^2$. Any such combinations can only be equal if the coefficients are the same, i.e. $a_\upsilon = \tilde{a}_\upsilon$ and $b_\upsilon = \tilde{b}_\upsilon$ if $a_\upsilon \neq 0$. Thus, for every $(i, j) \in [d+1]^2$, $\beta_{i,j} = \tilde{\beta}_{i,j}$ as $\alpha_{i,j}(\cdot) > 0$ almost surely. By bijection of the logistic function and using the fact that $w$ is arbitrary and the non-degeneracy assumption, $\gamma = \tilde{\gamma}$ and $\theta = \tilde{\theta}$.

Thus, we arrive at $\vartheta = \tilde{\vartheta}$, a contradiction. $\qquad\square$

# B  SIMULATION OF HAWKES PROCESSES

## B.1  SIMULATING HAWKES PROCESSES VIA OGATA'S THINNING ALGORITHM

The standard method for simulating Hawkes processes, known as *thinning* was introduced by Ogata (1981). In the univariate exponential case, with parameters $(\mu, \alpha, \beta) \in [0, \infty)^3$ suppose a trajectory has been simulated up to time $t \geq 0$, yielding a counting process $N : [0, t] \to \mathbb{N}$ with arrival times $(\tau_\ell)_{\ell=1}^{N(t)}$ and target intensity

$$\lambda = \mu + \sum_{\ell \in \mathbb{N}} \alpha \beta e^{-\beta(\cdot - \tau_\ell)} \, .$$

The thinning algorithm generates samples to continue the Hawkes process using rejection sampling, i.e. given an upper bound $\bar{\lambda} \geq \lambda$, it generates the next sample inter-arrival time according to a Poisson process with intensity $\bar{\lambda}$, with an acceptance probability of $\lambda(t)/\bar{\lambda}(t)$. This procedure can be repeated until a fixed time horizon $T \in [0, +\infty)$ or until a fixed number of samples is obtained.

In the multivariate case, the upper bound used is the sum of intensities over dimensions, i.e. $\bar{\lambda} = \sum_{i=1}^{d+1} \lambda_i$, and a sample accepted at time $t > 0$ is then randomly assigned to a dimension in proportion to $\lambda(t)/\bar{\lambda}(t)$. The resulting algorithm is presented in Algorithm 1.

---

**Algorithm 1** Multivariate Hawkes Simulation (Ogata's Thinning Method)

---

**Require:** $\mu \in [0, +\infty)^{d+1}, \alpha \in [0, +\infty)^{(d+1)^2}, \beta \in [0, +\infty)^{(d+1)^2}$, time horizon $T \in (0, +\infty]$, number of samples $\mathcal{N} \in \mathbb{N} \cup \{+\infty\}$

1: Initialize $t \leftarrow 0, n \leftarrow 0$ $\mathcal{H}_i \leftarrow [\,]$ for all $i$
2: Initialize $R_{i,j} \leftarrow 0$ for all $i, j$
3: **while** $t < T$ or $n \leq \mathcal{N}$ **do**
4:     Compute $\lambda(t) = \mu + \sum_j R_{\cdot,j}$
5:     Update $\bar{\lambda}(t) \leftarrow \sum_i \lambda_i(t)$
6:     Sample $s \sim \mathrm{Exp}(1/\bar{\lambda}(t))$
7:     Update $\lambda(t + s) \leftarrow \mu + \sum_j R_{\cdot,j} \circ \exp(-\beta_{\cdot,j} s)$
8:     Update $\bar{\lambda}(t + s) \leftarrow \sum_i \lambda_i(t + s)$
9:     Sample $u \sim \mathcal{U}(0, 1)$
10:    **if** $u \leq \bar{\lambda}(t + s)/\bar{\lambda}(t)$ **then**
11:        Random choose $\hat{j} \in [d + 1]$ with probability $\lambda(t + s)/\bar{\lambda}(t + s)$
12:        Append $t$ to $\mathcal{H}_{\hat{j}}$
13:        Update $n \leftarrow n + 1$
14:        **for** each $i \in \{1, \ldots, d + 1\}$ **do**
15:            $\nu_{i,\hat{j}} \leftarrow R_{i,\hat{j}} \, e^{-\beta_{i,\hat{j}} s} + \alpha_{i,\hat{j}} \beta_{i,\hat{j}}$
16:        **end for**
17:    **end if**
18:    Increment $t \leftarrow t + s$
19: **end while**
20: **return** $\{\mathcal{H}_i\}_{i=1}^{d+1}$

---

## B.2  IMPLEMENTATION IN FEATHAWKES

In `FeatHawkes`, Algorithm 1 is implemented using a residual memory matrix $R$ which is updated so that at any jump time $\tau$

$$R_{i,j}(\tau) = \sum_{\ell \in \mathbb{N}} \alpha_{i,j} \beta_{i,j} e^{-\beta_{i,j}(\tau - \tau_\ell^{(j)})} \mathbb{1}_{\{\tau_\ell \leq \tau\}}$$

where $(\tau_\ell^{(j)})_{\ell \in \mathbb{N}}$ denotes the sequence of jump times of dimension $j$. The value of $R$ is maintained online using

$$R(t + s) = R(t) \, e^{-\beta s}$$

when the global inter-arrival time $s \in [0, +\infty)$ is sampled by Algorithm 1. Using $R$, the simulation is performed efficiently by updating

$$\lambda(t) = \mu + \sum_{j=1}^{d+1} R_{\cdot,j}(t)$$

and using $\bar{\lambda} := \sum_{i=1}^{d+1} \lambda_i$ for importance sampling in Algorithm 1. The samples are allocated to a specific dimension $\hat{j}$ at random, which adds a new log to $R$ by

$$R_{\cdot,\hat{j}} \leftarrow R_{\cdot,\hat{j}} + \alpha_{\cdot,\hat{j}} \beta_{\cdot,\hat{j}} \,.$$

Throughout the simulation, event times and their corresponding types (dimensions) are stored in `self.times` and `self.types`. The process stops either once a given number of events have been generated or a specified time horizon is exceeded. An additional method, `_calculate_intensity`, can be used to compute $\lambda(t)$ across a discrete time grid for analysis or visualisation. For each evaluation time, it adds up the contribution from the baseline intensity and from all past events:

$$\lambda_i(t) = \mu_i + \sum_{j=1}^{d+1} \sum_{\tau_k^{(j)} < t} \alpha_{i,j} e^{-\beta_{i,j}(t - \tau_k^{(j)})}.$$

This implementation is designed to be efficient and scalable. By maintaining a compact memory matrix that is updated incrementally, it avoids the need to recompute full excitation sums at each time step. This makes the implementation of the algorithm well-suited for high-dimensional simulation tasks, such as those encountered in feature-based attribution models for player performance.

## C  IMPLEMENTATION OF HAWKES FITTING IN FEATHAWKES

### C.1  VECTORISED LIKELIHOOD CALCULATIONS

Let us consider the likelihood for $K$ trajectories $(N^{(k)})_{k=1}^{K}$ each on $[0, T_k]$ of an exponential Hawkes process, each respectively with arrival times $((\tau_\ell^{(k,i)})_{\ell=1}^{N^{(k)}(T_k)})_{i=1}^{d+1}$, that is

$$\mathcal{L}(N; \mu, \alpha, \beta) = -\sum_{k=1}^{K} \sum_{i=1}^{d+1} \sum_{\ell=1}^{N_i^{(k)}(T_k)} \log \left( \mu_i + \sum_{j=1}^{d+1} \sum_{m=1}^{\ell} \alpha_{i,j} \beta_{i,j} e^{-\beta_{i,j}(\tau_\ell^{(k,i)} - \tau_m^{(k,i)})} \right)$$

$$- \sum_{k=1}^{K} \sum_{i=1}^{d+1} \mu_i T_k + \sum_{j=1}^{d+1} \sum_{\ell=1}^{N_i^{(k)}(T_k)} \alpha_{i,j} \left( 1 - e^{-\beta_{i,j}(T_k - \tau_\ell^{(k,i)})} \right) . \tag{9}$$

The computation of the partial sums of the kernel in the first term are performed efficiently in `FeatHawkes` by computing a 3-tensor of differences $\Delta$ with

$$\Delta_{k,\ell,m} := t_\ell^{(k)} - t_m^{(k)} \,,$$

in which $(t_\ell^{(k)})_{\ell=1}$ is the sequence of jump times of $N^{(k)}$ (ignoring dimensions). Creating $\Delta$ is performed efficiently via `PyTorch` using tensor-specialised computation on GPUs. In order to conform the dimensions of $\Delta$, one pads it up to the maximum number of events across trajectories. In practical applications, the resulting additional computation still outperforms treating trajectories independently.

The tensor $\Delta$ contains some negative entries which do not appear in the first term of (9). These can be efficiently eliminated using an "upper-triangular" masking tensor $M$ defined by

$$M_{k,\ell,m} := \mathbb{1}_{\{m < \ell\}} \,.$$

Let $\delta$ denote the 2-tensor of dimensions, i.e. $\delta_{k,\ell} \in [d+1]$ gives the dimension in which a jump occurred at time $t_\ell^{(k)}$. Given coefficients $(\alpha, \beta)$, we can compute the 3-tensor of coefficients $\alpha$ effectively corresponding to each jump using

$$A_{k,\ell,m} := \alpha_{\delta_{k,\ell}, \delta_{k,m}}$$

and finally compute the first likelihood term from

$$\sum_{j=1}^{d+1}\sum_{m=1}^{\ell}\alpha_{i,j}\beta_{i,j}e^{-\beta_{i,j}(\tau_{\ell}^{(k,i)}-\tau_m^{(k,i)})}=[A\exp(-\beta\Delta\circ M)\circ M]_{k,\ell}$$

in which $\circ$ denote the element-wise (Hadamard) product of tensors, as

$$\Sigma(\log(\mu_{\delta}+A\exp(-\beta\Delta\circ M)\circ M))$$

in which $\Sigma$ denotes the sum of all entries of its tensor argument. In the presence of features, this computation continues to hold, up to making $\alpha$ dependent on $k$ as per (5).

The second term of the likelihood (9) can be computed efficiently in a similar manner.

## C.2 OPTIMISATION

In `FeatHawkes`, we fit exponential Hawkes processes by minimising a penalised likelihood of the form

$$\min_{\mu,\alpha,\beta}\mathcal{L}(N;\mu,\alpha,\beta)+\eta\Upsilon(\mu,\alpha,\beta)+\nu\left\|\alpha\right\|_1, \tag{10}$$

in which $(\eta,\nu)$ are positive meta-parameters and $\|\alpha\|_1:=\sum_{i,j}|\alpha_{i,j}|$ is the one norm of the flattened coefficient matrix. By default, $\nu=0$ (as we described in Section 3.1), but $\nu$ offers the possibility to regularise the coefficients in a parsimonious manner, which is desirable in some applications (Bacry et al., 2020; Zhou et al., 2013). In the presence of features, (10) is replaced by an analogue to (6), namely

$$\min_{\mu,\gamma,\theta,\beta}\mathcal{L}(N;\mu,\gamma,\theta,\beta|X,Z)-\eta\sum_{i=1}^{d+1}\log(\mu_i)+\sum_{j=1}^{d+1}\log(\beta_{i,j})+\nu(\|\theta\|_1+\|\gamma\|_1). \tag{11}$$

The minimisation of (10) and (11) is done in `FeatHawkes` using gradient-based methods. It is possible to use any optimiser implementable using `PyTorch`'s `optimizers` module. For the sake of simplicity, we present a stochastic gradient method. Setting a learning rate schedule $(\rho_s)_{s\in\mathbb{N}}$, at each step $s\in\mathbb{N}$, update

$$(\hat{\mu}_{s+1},\hat{\gamma}_{s+1},\hat{\theta}_{s+1},\hat{\beta}_{s+1})^{\top}=(\hat{\mu}_s,\hat{\gamma}_s,\hat{\theta}_s,\hat{\beta}_s)^{\top}-\rho_s\nabla\Gamma_s(\hat{\mu}_s,\hat{\gamma}_s,\hat{\theta}_s,\hat{\beta}_s),$$

wherein

$$\Gamma_s(\alpha,\gamma,\theta,\beta):=\sum_{k\in\Xi_s}^{K}\mathcal{L}(N^{(k)};\mu,\alpha(\gamma,\theta;X^{(k)},Z^{(k)}),\beta)$$

$$-\eta\sum_{i=1}^{d+1}\log(\mu_i)+\sum_{j=1}^{d+1}\log(\beta_{i,j})+\nu(\|\theta\|_1+\|\gamma\|_1)$$

for $(\Xi_s)_{s\in\mathbb{N}}$ an independent sequence of random subsets of $[K]$ each of size $\xi\in[K]$, i.e. mini-batches.

In terms of descent hyperparameters, `FeatHawkes` allows for early stopping criteria, for adjusting the learning rate schedule, including using classical heuristics such as `ReduceLROnPlateau`, and for choosing between a constant penalisation or simulated annealing by setting an adjustable sequence $(\eta_s)_{s\in\mathbb{N}}$ of log-barrier penalties.

## D COMPLEMENTS TO THE EXPERIMENTS

This appendix contains all complements to the experiments conducted in Section 4, ordered relative to the order of that section. Thus, we begin by reporting more detailed numerical results from synthetic data in Appendix D.1. Next, in Appendices D.3 and D.4, we describe the football match data used for the real-world experiments. In particular, the way we convert teams to vectors of dimensions in our Hawkes model and the features we used, respectively. In Appendix D.5, we mobilise this construction to give a demonstration of factual attribution for Liverpool, which we only summarised in Section 4. Using the same data, we perform a validation of the soundness of our exogenous features method in Appendix D.6. Finally, in Appendix D.7, we fully demonstrate the predictive attribution method on Kylian Mbappé's transfer from Paris Saint-Germain to Real Madrid.

### D.1 ADDITIONAL RESULTS FROM SYNTHETIC DATA

We provide in Tables 2 and 3 a detailed breakdown of the data of Figure 1.

Table 2: Numerical comparison of estimation error of our algorithm and EM.

| $T$ | $\|\tilde{\mu} - \hat{\mu}\|$ | $\|\tilde{\mu} - \hat{\mu}^{(\text{EM})}\|$ | $\|\tilde{\beta} - \hat{\beta}\|$ | $\|\tilde{\beta} - \hat{\beta}^{(\text{EM})}\|$ | $\|\tilde{\alpha} - \hat{\alpha}\|$ | $\|\tilde{\alpha} - \hat{\alpha}^{(\text{EM})}\|$ |
|---|---|---|---|---|---|---|
| 400 | 0.000441 | 0.000421 | 0.398946 | 0.588488 | 0.002788 | 0.002846 |
| 800 | 0.000125 | 0.000129 | 0.070473 | 0.058170 | 0.001087 | 0.001120 |
| 1200 | 0.000126 | 0.000129 | 0.053482 | 0.048189 | 0.000587 | 0.000663 |
| 1500 | 0.000088 | 0.000092 | 0.028381 | 0.029470 | 0.000516 | 0.000567 |
| 2000 | 0.000064 | 0.000062 | 0.018383 | 0.022729 | 0.000407 | 0.000455 |
| 3000 | 0.000056 | 0.000050 | 0.012951 | 0.015462 | 0.000329 | 0.000377 |
| 5000 | 0.000042 | 0.000029 | 0.004178 | 0.003363 | 0.000205 | 0.000253 |

Table 3: Numerical comparison of efficiency and overall error of our algorithm and EM.

| $T$ | Total error | EM Total error | Runtime | EM Runtime |
|---|---|---|---|---|
| 400 | 0.402175 | 0.591756 | 0.473366 | 1.000571 |
| 800 | 0.071686 | 0.059419 | 0.385347 | 4.825050 |
| 1200 | 0.054195 | 0.048981 | 0.559265 | 12.434479 |
| 1500 | 0.028984 | 0.030129 | 0.613456 | 18.865248 |
| 2000 | 0.018855 | 0.023246 | 0.622070 | 32.207077 |
| 3000 | 0.013336 | 0.015888 | 0.652892 | 75.800880 |
| 5000 | 0.004425 | 0.003646 | 0.752208 | 233.128337 |

Figure 4 presents the error in the estimation of the excitation ($\hat{\alpha}$) and descendent ($\hat{D}$) matrices of the Hawkes process described in (7) relative to the ground truth values ($\alpha, D$) as a function of the number of sample trajectories.

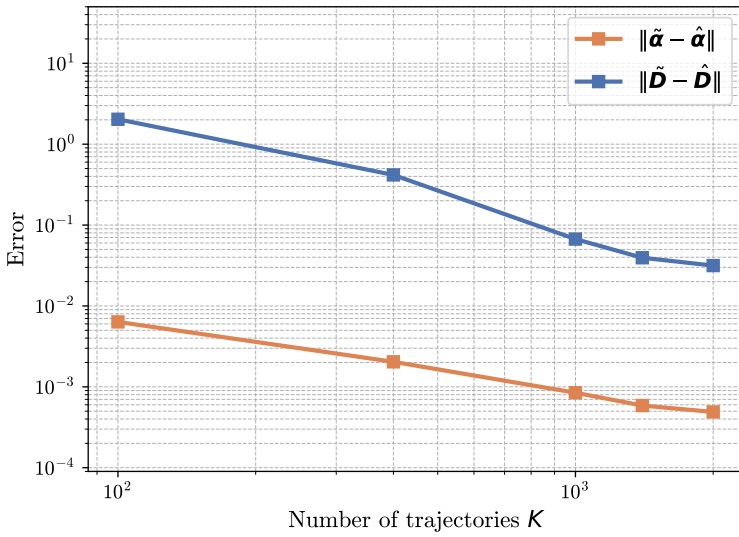

Figure 4: Estimation error for $(\hat{\alpha}, \hat{D})$ as a function of the number of trajectories $K$.

## D.2 FEATURE-BASED ATTRIBUTION ON SYNTHETIC FOOTBALL DATA

In this section we describe the experimental protocol used to compare different multi-touch attribution methods from the predicted last-touch attributions $\hat{\alpha}$ using the Hawkes model for the point-process.

In order to increase the realism of the experiment, we forgo the example of (7) in favour of a 12-dimensional simulation (matching the real-world football data from the next section). First, as before, we fix coefficient matrices

$$
\tilde{\mu} := \begin{pmatrix} 0.05000000 \\ 0.05909091 \\ 0.06818182 \\ 0.07727273 \\ 0.08636364 \\ 0.09545455 \\ 0.10454545 \\ 0.11363636 \\ 0.12272727 \\ 0.13181818 \\ 0.14090909 \\ 0.15000000 \end{pmatrix}, \quad
\tilde{\beta} := \begin{pmatrix} 0.6 \\ 0.6 \\ 0.6 \\ 0.6 \\ 0.6 \\ 0.6 \\ 0.6 \\ 0.6 \\ 0.6 \\ 0.6 \\ 0.6 \\ 0.6 \end{pmatrix}, \quad
\tilde{\gamma} := \begin{pmatrix}
-8 & -8 & -8 & -8 & -8 & -8 & -8 & -8 & -8 & -8 & -8 & -8 \\
-2 & -8 & -8 & -2 & -2 & -8 & -8 & -8 & -8 & -8 & -8 & -8 \\
-2 & -8 & -8 & -2 & -2 & -8 & -8 & -8 & -8 & -8 & -8 & -8 \\
-2 & -8 & -8 & -8 & -8 & -8 & -8 & -8 & -8 & -8 & -8 & -8 \\
-2 & -8 & -8 & -8 & -8 & -8 & -8 & -8 & -8 & -8 & -8 & -8 \\
-8 & -2 & -2 & -2 & -2 & -8 & -8 & -8 & -8 & -8 & -8 & -8 \\
-8 & -2 & -2 & -8 & -8 & -2 & -8 & -8 & -8 & -8 & -8 & -8 \\
-8 & -2 & -2 & -8 & -8 & -2 & -8 & -8 & -8 & -8 & -8 & -8 \\
-8 & -2 & -2 & -8 & -8 & -2 & -2 & -2 & -8 & -8 & -8 & -8 \\
-8 & -2 & -2 & -8 & -8 & -2 & -2 & -2 & -8 & -8 & -8 & -8 \\
-8 & -2 & -2 & -8 & -8 & -2 & -2 & -2 & -8 & -8 & -8 & -8 \\
-8 & -2 & -2 & -8 & -8 & -8 & -2 & -2 & -2 & -2 & -2 & -8
\end{pmatrix},
$$

$$
\tilde{\theta}^{(1)} := \begin{pmatrix}
0 & 0 & 0 & 0 & 0 & 0 & 0 & 0 & 0 & 0 & 0 & 0 \\
0.002 & 0 & 0 & 0.009 & 0.009 & 0 & 0 & 0 & 0 & 0 & 0 & 0 \\
0.002 & 0 & 0 & 0.009 & 0.009 & 0 & 0 & 0 & 0 & 0 & 0 & 0 \\
0.002 & 0 & 0 & 0 & 0 & 0 & 0 & 0 & 0 & 0 & 0 & 0 \\
0.002 & 0 & 0 & 0 & 0 & 0 & 0 & 0 & 0 & 0 & 0 & 0 \\
0 & 0.18 & 0.18 & 0.009 & 0.009 & 0 & 0 & 0 & 0 & 0 & 0 & 0 \\
0 & 0.18 & 0.18 & 0 & 0 & 0.2925 & 0 & 0 & 0 & 0 & 0 & 0 \\
0 & 0.18 & 0.18 & 0 & 0 & 0.2925 & 0 & 0 & 0 & 0 & 0 & 0 \\
0 & 0.18 & 0.18 & 0 & 0 & 0.2925 & 0.48 & 0.48 & 0 & 0 & 0 & 0 \\
0 & 0.18 & 0.18 & 0 & 0 & 0.2925 & 0.48 & 0.48 & 0 & 0 & 0 & 0 \\
0 & 0.18 & 0.18 & 0 & 0 & 0.2925 & 0.48 & 0.48 & 0 & 0 & 0 & 0 \\
0 & 0.09 & 0.09 & 0 & 0 & 0 & 0.48 & 0.48 & 0.95 & 1.25 & 0.95 & 0
\end{pmatrix},
$$

$$
\tilde{\theta}^{(2)} := \begin{pmatrix}
0 & 0 & 0 & 0 & 0 & 0 & 0 & 0 & 0 & 0 & 0 & 0 \\
0.054 & 0 & 0 & 0.081 & 0.081 & 0 & 0 & 0 & 0 & 0 & 0 & 0 \\
0.054 & 0 & 0 & 0.081 & 0.081 & 0 & 0 & 0 & 0 & 0 & 0 & 0 \\
0.018 & 0 & 0 & 0 & 0 & 0 & 0 & 0 & 0 & 0 & 0 & 0 \\
0.018 & 0 & 0 & 0 & 0 & 0 & 0 & 0 & 0 & 0 & 0 & 0 \\
0 & 0.324 & 0.324 & 0.162 & 0.162 & 0 & 0 & 0 & 0 & 0 & 0 & 0 \\
0 & 0.486 & 0.486 & 0 & 0 & 0.5265 & 0 & 0 & 0 & 0 & 0 & 0 \\
0 & 0.486 & 0.486 & 0 & 0 & 0.5265 & 0 & 0 & 0 & 0 & 0 & 0 \\
0 & 0.756 & 0.756 & 0 & 0 & 0.819 & 1.008 & 1.008 & 0 & 0 & 0 & 0 \\
0 & 1.026 & 1.026 & 0 & 0 & 1.1115 & 1.368 & 1.368 & 0 & 0 & 0 & 0 \\
0 & 0.756 & 0.756 & 0 & 0 & 0.819 & 1.008 & 1.008 & 0 & 0 & 0 & 0 \\
0 & 0 & 0 & 0 & 0 & 0 & 0 & 0 & 0 & 0 & 0 & 0
\end{pmatrix}.
$$

Next, we generate features $(\tilde{X}'_i)_{i \in [d+1]}$ (again foregoing $Z$ for simplicity) according to Gaussian distributions whose means are chosen to mimic key properties of football match data. We structure excitation coefficients to mimic real football data by imposing a sparse adjacency pattern that reflects plausible ball progression, enforcing strong negative biases on forbidden links, and prohibit any outgoing effect from the surface node ($\tilde{\theta}^{(2)}_{12} = 0$). Weights are scaled by role, with larger values assigned to attacking channels, moderate values to midfield relays, and smaller values to defenders. Finally, all coefficients are tuned so that the resulting Hawkes matrix remains *subcritical* (spectral radius $< 1$), ensuring stability and a well-defined branching operator. The complete generation procedure is available in `FeatHawkes`.

From the $\kappa' = 150$ sampled copies of $(\tilde{X}'_i)_{i \in [d+1]}$ and the coefficients $(\tilde{\gamma}, \tilde{\theta})$, we generate $K' = \kappa'$ trajectories and fit our Hawkes model to these, obtaining the direct (first-touch) attribution $\hat{\alpha}$ under the Hawkes model. This experiment aims to compare different methods for performing indirect (multi-touch) attribution from this point-process fit. On the one hand, for our method, we compute the empirical descendant matrix $\hat{D}$. On the other hand, for $j \in [d+1]$, let $\hat{\alpha}^{(j)}$ denote the matrix $\hat{\alpha}$ with the row $j$ and column $j$ removed, from which one computes $\hat{D}^{(j)} := \hat{\alpha}^{(j)} * (Id - \hat{\alpha}^{(j)})^{-1}$ the attribution in absence of player $j$ and $\hat{s}^{(j)} := \sum_{i \in [d+1]} \hat{D}^{(j)}_{12,i}$ the total value generated, which yields their uplift $U_j$ via

$$U_j = \hat{s} - \hat{s}^{(j)}$$

wherein $\hat{s} := \sum_{i \in [d+1]} \hat{D}_{12,i}$. Finally, we compute attribution scores using Shapley values by extending the uplift methodology by denoting by $\hat{\alpha}^{(S)}$, where $S \subset [d]$, the analogue of $\hat{\alpha}^{(j)}$ in which all rows and columns whose indices are in $S$ have been deleted. The Shapley values are then computed combinatorially using

$$S_j := \sum_{S \in \{\mathcal{S} \in \mathcal{P}([d]) : j \in \mathcal{S}\}} \frac{(d - |S|)!(|S| - 1)!}{d!} \left( s^{(S)} - s^{(S \setminus \{j\})} \right)$$

wherein $\mathcal{P}([d])$ denotes the power set of $[d]$.

We present in Table 4 the resulting estimated values for each method, as well as the errors in each dimension. Note that, in order to facilitate comparisons across methods, Shapley values and Uplift are renormalised to sum to 1 for ease of comparison with our method.

Table 4: Comparison of indirect attribution heads on top of $\hat{\alpha}$. **Dim.** denotes the number of dimension, **GT** denotes ground truth, i.e. $D_{12,i}$, **Ours** denotes $\hat{D}_{12}$, **Shapley**, $S_j$, and **Uplift**, $U_j$. Each "error" column contains the difference of **GT** and the value of the method to the left of it. The last line contains the means of the sums of squares of the "error" columns.

| Dim. | GT | Ours | Error | Shapley | Error | Uplift | Error |
|---|---|---|---|---|---|---|---|
| 1 | 0.000560 | 0.017406 | 0.016847 | 0.017314 | 0.016754 | 0.020033 | 0.019473 |
| 2 | 0.042158 | 0.057407 | 0.015249 | 0.077540 | 0.035382 | 0.083067 | 0.040909 |
| 3 | 0.000627 | 0.047148 | 0.046522 | 0.049066 | 0.048440 | 0.052012 | 0.051385 |
| 4 | 0.000627 | 0.027246 | 0.026619 | 0.024980 | 0.024353 | 0.030449 | 0.029823 |
| 5 | 0.015112 | 0.048843 | 0.033731 | 0.049218 | 0.034105 | 0.065204 | 0.050092 |
| 6 | 0.042158 | 0.058239 | 0.016081 | 0.076030 | 0.033872 | 0.082443 | 0.040285 |
| 7 | 0.101764 | 0.096691 | -0.005073 | 0.110046 | 0.008282 | 0.112707 | 0.010943 |
| 8 | 0.218193 | 0.156551 | -0.061643 | 0.130437 | -0.087756 | 0.121531 | -0.096662 |
| 9 | 0.100839 | 0.097653 | -0.003186 | 0.108224 | 0.007385 | 0.111818 | 0.010979 |
| 10 | 0.223901 | 0.176266 | -0.047634 | 0.152849 | -0.071051 | 0.141122 | -0.082778 |
| 11 | 0.254061 | 0.216548 | -0.037514 | 0.204296 | -0.049766 | 0.179614 | -0.074448 |
| **MSE** | | **0.001118** | | 0.002012 | | 0.002881 | |

Inspection of Table 4 reveals that all methods perform similarly, successfully recovering the important dimensions of the ground truth, but that the magnitude of the means squared error of our method is 50% lower, validating the use of the descendant matrix even in the presence of estimation noise.

### D.3 ASSIGNMENT OF PLAYERS TO DIMENSIONS

Football teams can adopt a range of compositions which modify how players are arranged on the field (for some examples, see Figure 5). One must assume that different compositions will improve or diminish the effectiveness of some players, *ceteris paribus*, and thus affect the regression coefficients of our model. One could fit the model player-by-player, encoding the position as a feature, but this removes the statistical basis for predictive evaluation of transfers. Instead, we will assign the dimensions to tactical positions using a methodology which remains consistent across the majority of

team compositions (*circa* 70% in our data), namely 4-2-3-1 (see Figure 5a) and 4-3-3 (see Figure 5b) compositions, and discard the remaining data. One can, of course, fit separate models for these discarded formations, but, for simplicity, we will focus on teams with stable and similar formations with four defenders.

We will use data spanning the 2023–2024 and 2024–2025 seasons. To ensure the sanity of our dataset $(N^{(k)})_k$, we use only teams which retained a 4-3-3 or 4-2-3-1 formation over the season, and we stop logging a match for a team when the team makes a substitution. If the new team stays stable for at least ten minutes, we use the resulting play data to start a new "match".

To assign dimensions to players, we start allocating from the back to the front and from left to right according to five general groups: the goal (who receives dimension #1), the defenders (who receive dimensions #2–5 from the left rear to the right rear), the defensive midfielders, the forward midfielders, and the striker (if applicable). This system is designed for 4-2-3-1 and 4-3-3 teams, as the same dimensions play a similar tactical role in both. In Figure 5c and Figure 5d we show how it can be extended to 4-4-2 and 4-3-2-1 configurations in a similar way, though we discard these uncommon formations are the numbering is a less good fit.

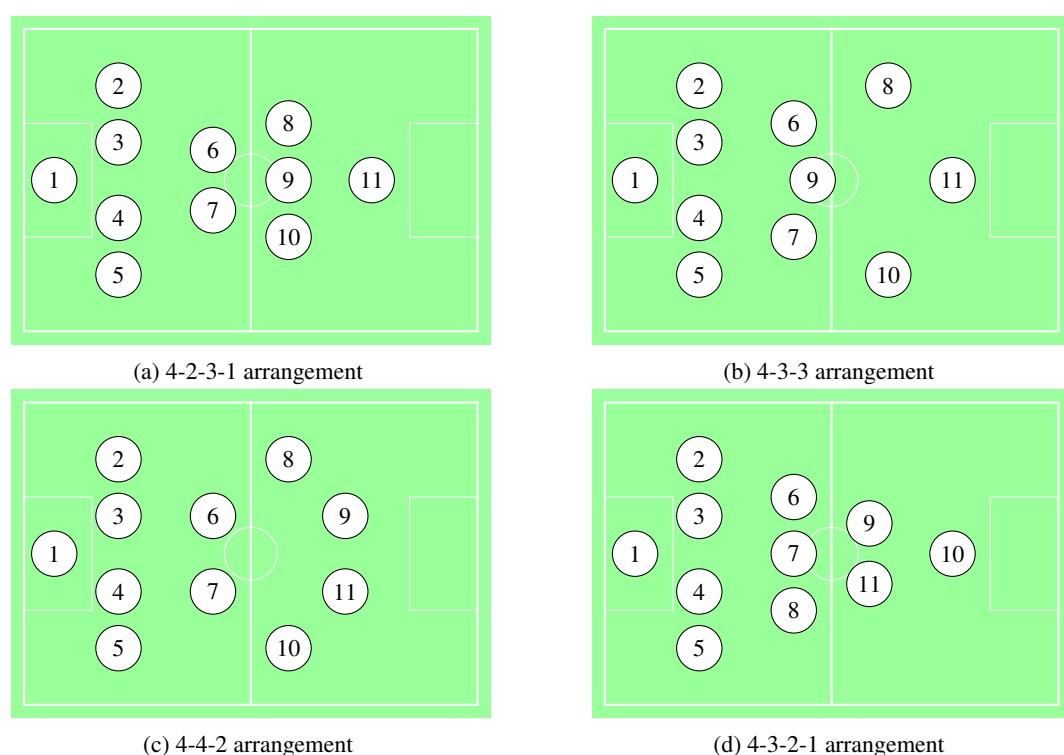

(a) 4-2-3-1 arrangement      (b) 4-3-3 arrangement

(c) 4-4-2 arrangement      (d) 4-3-2-1 arrangement

Figure 5: Common football formations (opposition to the right), players are numbered from 1 (goalkeeper) to 11 (rightmost forward) according to our procedure.

Recall that dimension 12 corresponds to the outcome, which is either the entry of the ball in the penalty box of the other team, or a strike estimated to have at least 25% probability of being on target according to a proprietary prediction model developed by Footovision, a sports analytics firm. Choosing a constructed feature instead of just scored goals is standard practice in sports analytics (Baouan et al., 2023) as it helps with sparsity of signals and smooths out random and adversarial factors that determine the conversion of on-target shots to goals.

### D.4 FEATURE DATA USED FOR THE PREDICTIVE ANALYSIS

The features we used are proprietary to and constructed by Footovision, a sports analytics firm, by aggregating match-level data. These features are engineered and selected with specialist expertise for their relevance to data-driven decision-making by football managers. In this example, we use only one team-wide feature $Z^{(k)}$, which is an indicator of the kind of formation used in the match (4-3-3 or 4-2-3-1).

At the player level, in contrast, we used a range of features. All features using match-data were calculated on the whole 2023 calendar year, while we will use only data from the second half of the 2023–2024 season for the Hawkes model. This avoids any feedback loop between the features and the Hawkes process. In total, the features used were:

- **Height and age**: height in centimetres measured at the start of the season, and age in years on the 1$^{\text{st}}$ July 2023.

- **Playtime**: average share of the total match time spent on the pitch by the player, expressed in minutes between 0 and 90.

- **Distance covered**: average total amount of ground covered, expressed per 90 minutes.

- **Movements received**: average number of successful ball receptions when the receiver had moved before receiving, expressed per 90 minutes.

- **One-touch passes**: average number of passes completed immediately after receiving the ball (without any controlling touch ) expressed per 90 minutes.

- **Forward line-breaking passes** : average number of completed forward passes that penetrated at least one opposition defensive line, expressed per 90 minutes.

- **Crosses excluding penalty area**: average number of crosses delivered from the wide channels (lateral distance of at least 20 meters from the pitch centre) that landed outside the penalty area, expressed per 90 minutes.

- **Passes leading directly to shots**: average number of passes by the player which immediately led the receiver to shoot, expressed per 90 minutes.

- **Offensive duels won** : average number of one-versus-one situations initiated by the attacking player (dribbles, shoulder-to-shoulder duels) that result in maintained possession or drawing a foul, expressed per 90 minutes.

- **Passed players with ball drives**: average number of "overpassed" players, expressed per 90 minutes.

- **VAEP (Valuing Actions by Estimating Probabilities)**: Total VAEP generated, expressed per 90 minutes. VAEP is a framework for valuing player actions in football. It assigns a value to each on-the-ball action in based on its impact on the game outcome while accounting for the context in which the action happened (Decroos et al., 2019). An action value reflects the action's expected influence on the scoreline. That is, an action valued at +0.05 is expected to contribute 0.05 goals in favour of the team performing the action, while an action valued at -0.01 is expected to yield 0.01 goals for their opponent.

As we offer $L_1$ regularisation of the estimated regression coefficients $\theta$, see (10), we renormalise all the features to have zero mean and unit variance. This also ensures that the magnitude of coefficients is comparable across features.

### D.5 ILLUSTRATION OF THE ATTRIBUTION METHOD ON LIVERPOOL FC'S 2023–2024 SEASONS

We illustrate the estimated excitation coefficients of the featureless model in Figure 6 using the data from the 2023-2024 season of Liverpool FC (we provide a summary of the team in Table 5). In this figure, the matrix $\alpha$ is represented as a heatmap, on which darker colour indicates a stronger excitation effect. Recall that $\alpha_{i,j}$ is the excitation coefficient from player $j$ to player $i$, so that, for

instance, Van Dijk (#3, the third line in Figure 6) receives a strong excitation effect from Robertson (#2, second column) and from Konaté (#4, fourth column), who stand next to him on the field. The apparent cluster in the top left corner of the matrix corresponds to the consistent passing patterns of defenders and the goalkeeper moving the ball around the back of the field, while the last line shows clearly the importance of the lateral attacking midfielders, Gakpo (#8) and Salah (#10) and the forward Diaz (#11) in creating direct outcomes.

Table 5: Team Composition (4-2-3-1) of Liverpool FC in the 2023-2024 season.

| Dimension # | Jersey # | Role | Player name |
|---|---|---|---|
| 1 | 1 | Goal Keeper | Alisson |
| 2 | 26 | Left Back | Robertson |
| 3 | 4 | Left Centre Back | Van Dijk |
| 4 | 5 | Right Centre Back | Konaté |
| 5 | 66 | Right Back | Alexander-Arnold |
| 6 | 10 | Left Defensive Midfielder | Mac Allister |
| 7 | 38 | Right Defensive Midfielder | Gravenberch |
| 8 | 18 | Left Attacking Midfielder | Gakpo |
| 9 | 8 | Center Attacking Midfielder | Szoboszlai |
| 10 | 11 | Right Attacking Midfielder | Salah |
| 11 | 7 | Striker | Diaz |
| 12 | - | *Outcome* | - |

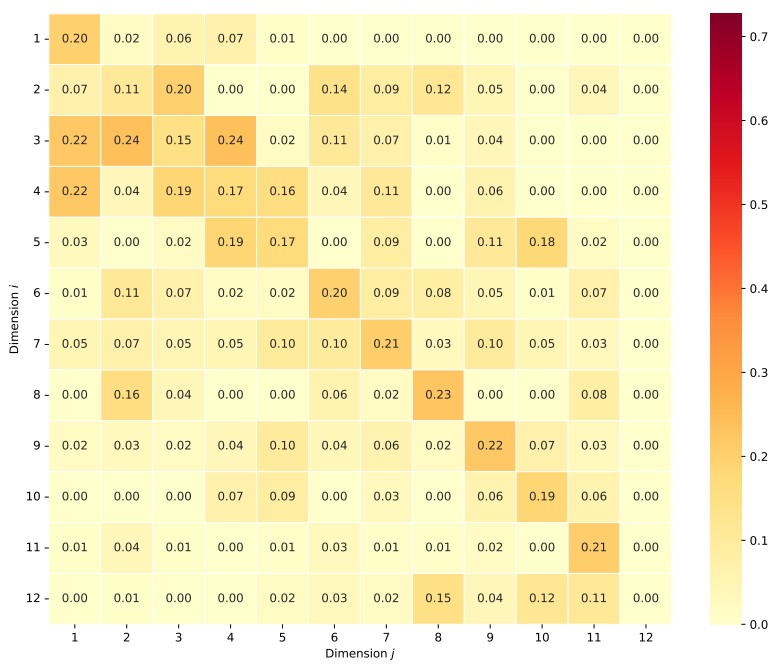

Figure 6: Estimated coefficient matrix $\hat{\alpha}$ for Liverpool FC in the 2023-2024 season.

While it only corresponds to last-touch attribution, the matrix $\alpha$ is already insightful for practitioners. Some insights easily seen are that Gakpo (#8, the left attacking midfielder) generates the most direct outcomes of the whole team, more than the other two attacking midfielders (Szoboszlai #9 and Salah #10), and over 30% more than the nominal striker Diaz (#11). Diaz (#11) appears quite isolated at the tip of the team, as the eleventh line of the matrix is almost entirely null (indicating he barely gets served by his teammates outside of the penalty box). This is likely because when he does get the ball, he is already in the penalty box and censored by the data-construction procedure. Nevertheless, he must be quite an effective player as his direct effect $\hat{\alpha}_{12,11} = 0.11$ is still significant.

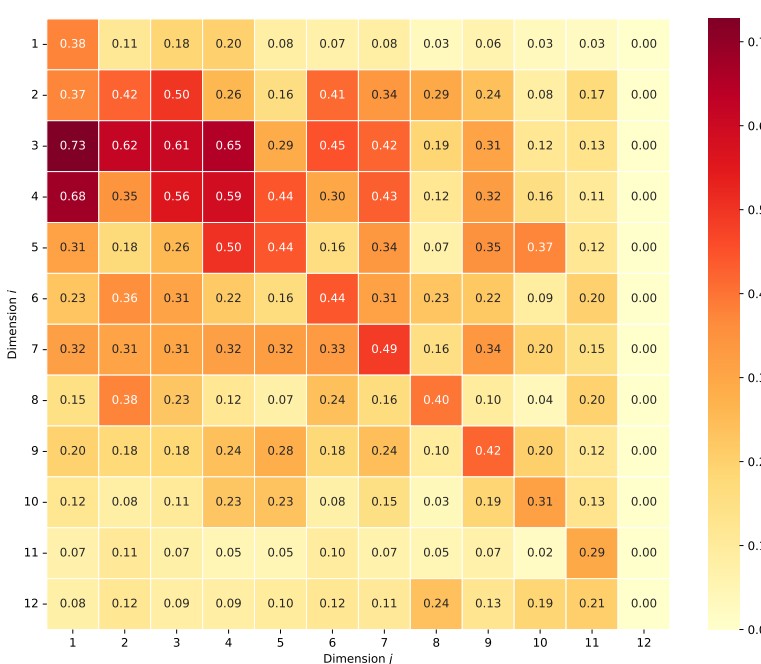

Figure 7: Estimated descendents matrix $\hat{D}$ for Liverpool FC in the 2023-2024 season.

In contrast, Figure 7 shows the (scaled) estimated descendents matrix $\hat{D} := \hat{\alpha}(I - \hat{\alpha})^{-1}$ for the same data, i.e. the multi-touch attribution of the outcomes. Recall that the descendents matrix encodes the attribution of the outcomes to the players, including both direct and indirect effects. Comparing Figure 6 and Figure 7, we can see that Diaz (#11) nearly doubles his expected contribution to 0.21, overtaking Salah (#10) at 0.19. Similarly, we can see the outsized contribution of Robertson (#2) amongst defenders, making him overall as important as the rear and central midfielders for the team's overall offence. Gakpo (#8) remains the most valued player overall.

A good overall picture of the team dynamics and attribution can be obtained by visualising the players according to both their overall importance and their excitation coefficients, which visualise the movement of the ball between players, as demonstrated by Figure 8. In this figure, the cycle of defenders identified in Figure 6 is clearly visible, and we can clearly see Robertson's (#2) preference for passing far down the left side to Gakpo (#8) who poses a direct threat to the other team, which drives his own importance up. Szoboszlai (#9), unlike the other attacking midfielders (Gakpo, #8, and Salah, #10), is a major ball distributor, as are the defending midfielders Mac Allister (#6) and Gravenberch #7. However, examining the excitation graph around the midfield highlights the profound left-right asymmetry in the team dynamics: Szoboszlai (#9) faces squarely right, as he and Gakpo (#8) do not interact at all, the latter relying on service directly from Robertson (#2) in his rear.

### D.6 FEATURE-BASED ATTRIBUTION ON REAL-WORLD DATA FROM THE 2023–2025 SEASONS

In this section, we present some further details on the empirical validation of the feature-based Hawkes model described in Section 4.2. We can separate (non-team) effects of the pair $(i, j)$ into three categories: the receiver effects $(\theta_{i,j}^{(1)})$, which depend on the features of $i$, emitter effects $(\theta_{i,j}^{(2)})$ dependent on the features of $j$, which, when $i, j$ combine into self-excitation effects $(\theta_{i,i}^{(u)}, u \in \{1, 2\})$.

A selection of other insightful receiver effects includes:

- **Playtime**, when $j$ is a striker and $i = d + 1$ is the outcome. The more a striker is present, as a share of the team's play time, the stronger the excitation they exert on the outcome

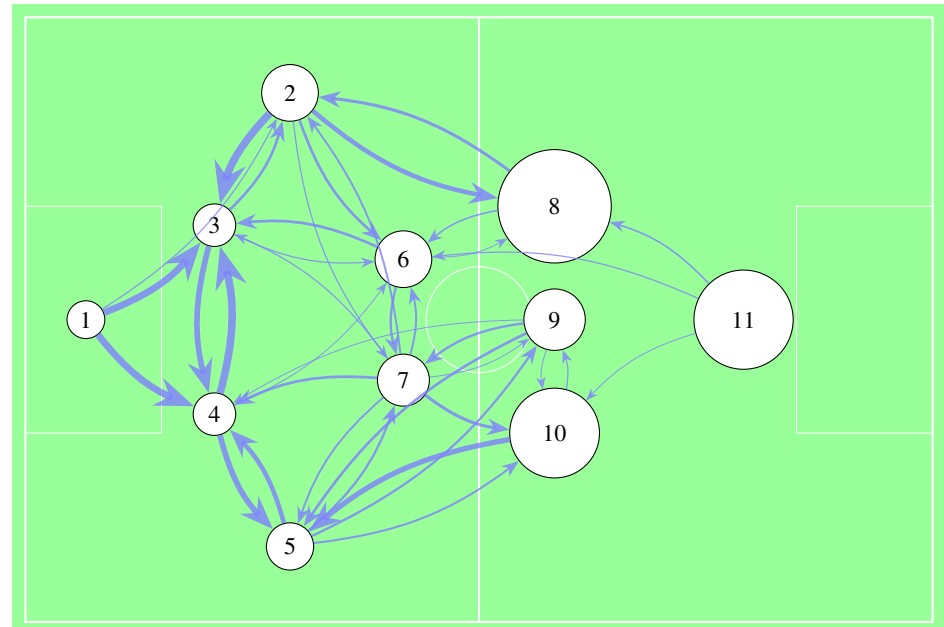

Figure 8: Liverpool's 4-2-3-1 formation for the 2023–2024 season, with players numbered according to Table 5. The size of each player is proportional to their importance as estimated by $\hat{D}_{12,\cdot}$ with our method, the thickness of the arrow from $j$ to $i$ is proportional to the estimated excitation effect $\hat{\alpha}_{i,j}$ (values smaller than 0.05 are not shown).

dimension, with $\hat{\theta}_{12,11}^{(1)} = 0.84$. Two interpretations appear: first, being a better striker is likely to lead one to be put on the pitch more and also to trigger more outcomes. Second, it supports the adaptation to a tactical role (rhythm of play, team offensive focus) over time.

- **Distance covered**, when $j$ is a left winger and $i$ is striker. Higher movement is associated with increased excitation forwards ($\hat{\theta}_{i,j}^{(1)} = 0.54$), underscoring the spatial importance of off-ball movements in attracting the ball and pushing it to the forefront of the team.

- **Offensive duels won**, when $j$ is a right winger and $i$ is a striker. Successful one-on-one actions by the winger significantly increase the likelihood of a pass to the striker ($\hat{\theta}_{i,j}^{(1)} = 0.63$), capturing the archetypal wing-to-strike transition following a dribble or cross.

On the emitter side, we can identify:

- **One-touch passes** when $j$ is a striker and $i$ is a neighbouring player. Strikers prone to one-touch passes substantially increase their excitation towards midfielders and wingers ($\hat{\theta}_{i,j}^{(2)} = 0.43$), indicative of quick combination play and short build-up sequences around the penalty box.

- **Movement received**, when $j$ is a left central midfielder and and $i$ is the outcome. Midfielders' off-ball movement has a strong positive effect on excitation towards other offensive units ($\hat{\theta}_{i,j}^{(2)} = 0.76$), highlighting a central orchestrating role facilitating spatial availability.

- **Crosses excluding penalty area**, when $j$ is a right winger and $i$ is a striker. The positive excitation coefficient $\hat{\theta}_{i,j}^{(2)} = 0.62$ reinforces the intuition that there is a stable tactical pathway between the right wing and the striker, reflective of structured offensive transitions.

When $i = j$, the features for both players are the same, and we report only the average estimated coefficient $\hat{\theta}_{i,i} := (\hat{\theta}_{i,i}^{(1)} + \hat{\theta}_{i,i}^{(2)})/2$. This coefficient is the change (in log-odds scale) of the magnitude of excitation a player has on themselves, i.e. on how many ball touches in a row they complete before passing or losing the ball. These coefficients are quite dependent on positions:

- Strikers have positive coefficients for **one-touch passes** ($\hat{\theta}_{i,i} = 0.13$) and for **offensive duels won** ($\hat{\theta}_{i,i} = 0.12$, which suggest that these events often lead to rapid repetitions, such as shots following rebounds or quick follow-ups.

- Central midfielders have a negative coefficient ($\hat{\theta}_{i,i} = -0.56$) with **passed players with ball drives** suggesting that players who excel at dribbling past other players tend to prefer to redistribute the ball when they have it, likely to relieve pressure or rotate play.

Overall, these results support the existence of stable and interpretable patterns in inter-player interactions. They provide valuable insight into tactical rhythms and the emergent structure of team dynamics, both of which are critical for performance optimisation and strategic planning.

The model was fitted using the procedure of Appendix C with the standard ADAM algorithm implemented in `PyTorch` and the following hyperparameters: a base learning rate of $\rho_0 = 0.005$, a boundary penalisation $\eta = 0.5$, an $\ell_1$ penalisation $\nu = 0.3$, and an improvement stopping criterion of $0.001$. In addition, we use the `ReduceLROnPlateau`, by decreasing the new learning rate by a factor of $0.8$ whenever the loss fails to improve by more than $0.2$ for $4$ epochs in a row.

### D.7    PREDICTIVE ATTRIBUTION ON FOOTBALL DATA

Before applying predictive attribution, we use our factual attribution method (see Appendix D.5) to evaluate the coefficient and descendant matrices $\hat{\alpha}^{\mathrm{PSG}}$ and $\hat{D}^{\mathrm{PSG}}$ for PSG in 2023–2024, when Kylian Mbappé was part of the team. We show these matrices in Figure 9 and Figure 10 respectively. That season, PSG played mostly in a 4-2-3-1 configuration with Mbappé as a central attacking midfielder (occasionally swapping with the left attacking midfielder Dembele, who we number #8), i.e. in dimension #9 according to Figure 5a.

Inspecting these matrices allows us to identify the playstyle of Mbappé in Paris: he is a highly influential player ($\hat{D}^{\mathrm{PSG}}_{12,09} = 0.30$) who generates a significant amount of threat directly ($\hat{\alpha}^{\mathrm{PSG}}_{12,9} = 0.13$). Looking at who serves him, i.e. the ninth line of $\hat{\alpha}^{\mathrm{PSG}}$, we see that he relies heavily on Hakimi (#2, deep in his rear) and Dembele (#8, on his left) to serve him.

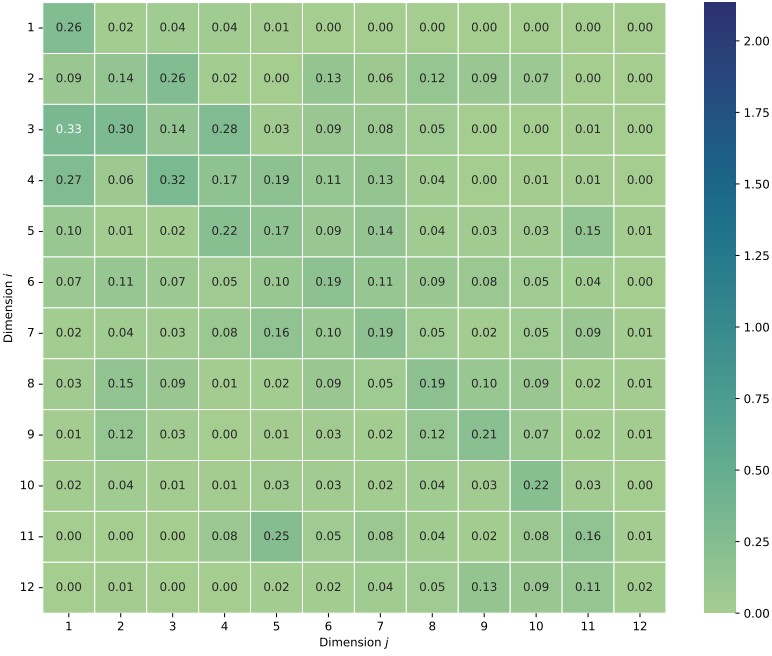

Figure 9: Estimated coefficients matrix $\hat{\alpha}^{\mathrm{PSG}}$ for PSG in the 2023–2024 season.

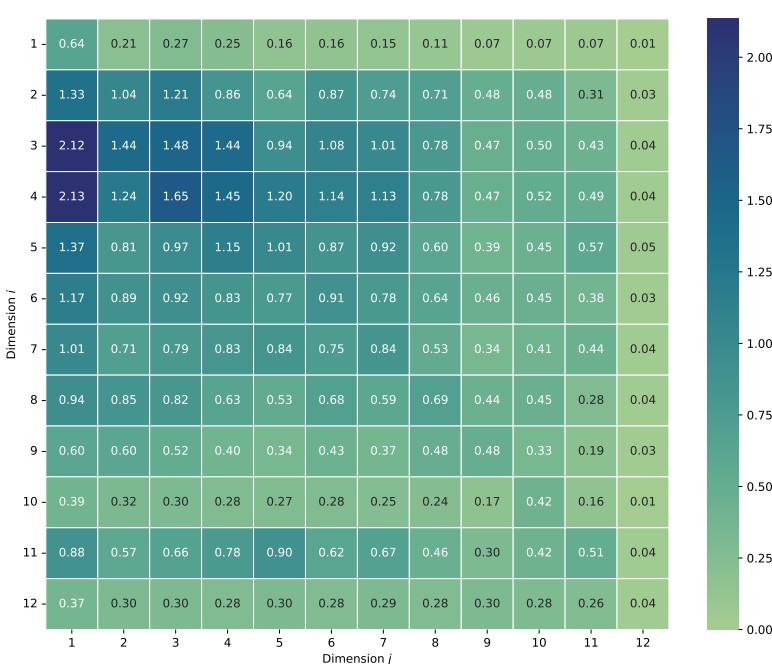

Figure 10: Estimated descendents matrix $\hat{D}^{\mathrm{PSG}}$ for PSG in the 2023–2024 season.

Likewise, we use our factual attribution method to compute the actual team influences ($\hat{\alpha}^{\mathrm{RM}}$ and $\hat{D}^{\mathrm{RM}}$) on Real Madrid's 2024–2025 season, which we use as a ground truth for the predictions. Real Madrid played a 4-2-3-1 configuration that season (see Figure 5a), meaning Mbappé retains the number #9 in the predictive scenario and in his actually played season at Real Madrid. Despite the disappointing performance of the team as a whole that year, Mbappé's direct influence on the outcome variable was evaluated at $\hat{\alpha}^{\mathrm{RM}}_{12,9} = 0.10$ and his overall influence at $\hat{D}^{\mathrm{RM}} = 0.17$, as shown on Figures 11 and 14, respectively.

Using the trained predictive model, we can evaluate alternative team compositions by using the features of the hypothetical considered team. As an illustration, we insert Kylian Mbappé's feature vector into dimension (#9) of Real Madrid's 2024–2025 line-up and compute the associated estimated Hawkes coefficients, and in turn the branching and descendant matrices $\hat{\alpha}^{\mathrm{CF}}$ and $\hat{D}^{\mathrm{CF}}$. We visualise these matrices as heatmaps in Figure 11.

While the model gives outright predictions of Mbappé's influence ($\hat{\alpha}^{\mathrm{CF}}_{12,9} = 0.05$ and $\hat{D}^{\mathrm{CF}}_{12,9} = 0.17$), it also gives a prediction of team dynamics as a whole which we can read from Figure 13. Indeed, the model appears to predict that Mbappé (#9) would behave more as a *reactive finisher* than an initiator, meaning he is more likely to be on the final receiver of the ball before the outcome rather than the initiator of a sequence to the outcome. This is evidenced by comparing his incoming branching ratio $\sum_j \hat{\alpha}^{\mathrm{CF}}_{9,j} = 0.65$ (with $\sum_{j \neq 9} \hat{\alpha}^{\mathrm{CF}}_{9,j} = 0.47$ coming from team-mates) to his outgoing branching ratio $\sum_i \hat{\alpha}^{\mathrm{CF}}_{i,9} = 0.45$ (of which only 0.27 goes to other players). In other words, Mbappé receives more than he distributes, marking him as a textbook finisher. The model estimates a pronounced self–excitation coefficient ($\hat{\alpha}^{\mathrm{CF}}_{9,9} = 0.18$), meaning that roughly 18% of Mbappé's actions would trigger another event involving him, suggesting he is capable of developing his flank on his own.

The model predicts that powerful precursors of Mbappé will be García (#2) with $\hat{\alpha}^{\mathrm{CF}}_{9,2} = 0.16$, Vinicius (#10) with $\hat{\alpha}^{\mathrm{CF}}_{9,10} = 0.14$, and Valverde (#8) with $\hat{\alpha}^{\mathrm{CF}}_{9,8} = 0.10$. In contrast, the players it believes he will most often excite are Vincius (#10) with $\hat{\alpha}^{\mathrm{CF}}_{10,9} = 0.07$, Valverde (#8) with $\hat{\alpha}^{\mathrm{CF}}_{8,9} = 0.06$, and Rodrygo (#11) with $\hat{\alpha}^{\mathrm{CF}}_{11,9} = 0.05$. Hence, the model's prediction highlights a preferred attacking pathway for the team, namely

$$\text{García/Vinicius/Valverde} \longrightarrow \text{Mbappé} \longrightarrow \text{Valverde/Vinicius/Rodrygo} \longrightarrow \text{Outcome,}$$

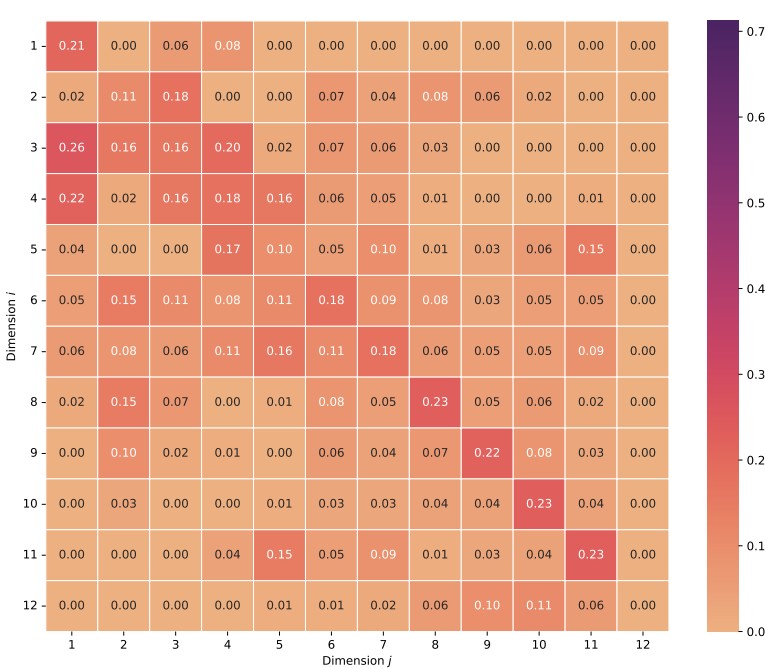

Figure 11: Estimated coefficients matrix $\hat{\alpha}^{\mathrm{RM}}$ for RM in the 2024–2025 season.

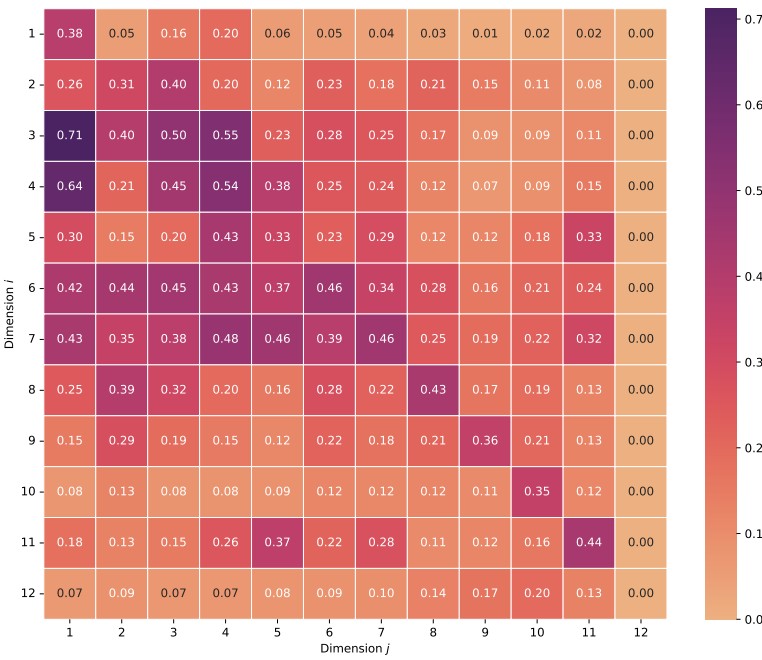

Figure 12: Estimated descendents matrix $\hat{D}^{\mathrm{RM}}$ for RM in the 2024–2025 season.

underlining Mbappé's role as a high-frequency finisher who rapidly recycles possession but is predominantly activated by wide or defensive recoveries.

The predictive model correctly predicted the overall influence of Mbappé at Real Madrid, with $\hat{D}^{\mathrm{CF}}_{12,9} = \hat{D}^{\mathrm{RM}12,9} = 0.17$, but it failed to accurately predict immediate influence (attribution) as

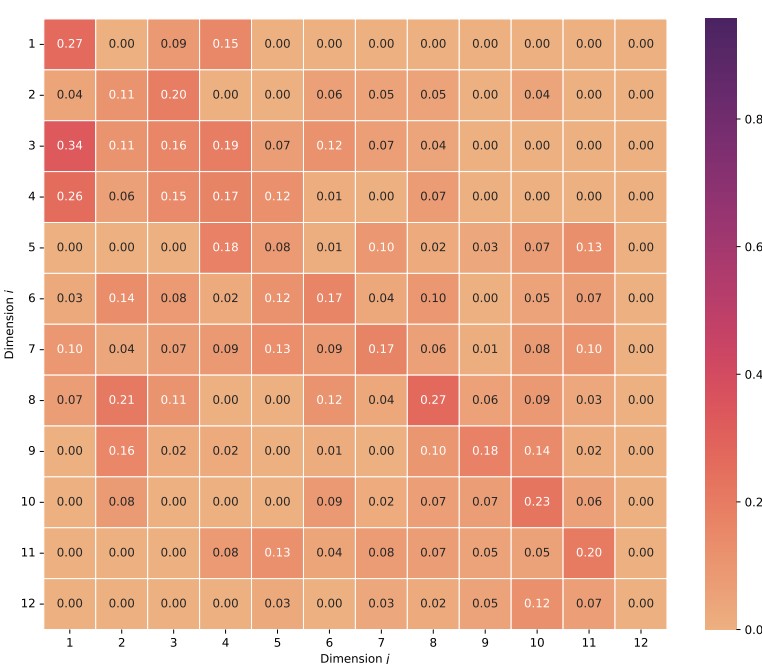

Figure 13: Predicted coefficients matrix $\hat{\alpha}^{\text{CF}}$ for RM in the 2024–2025 season.

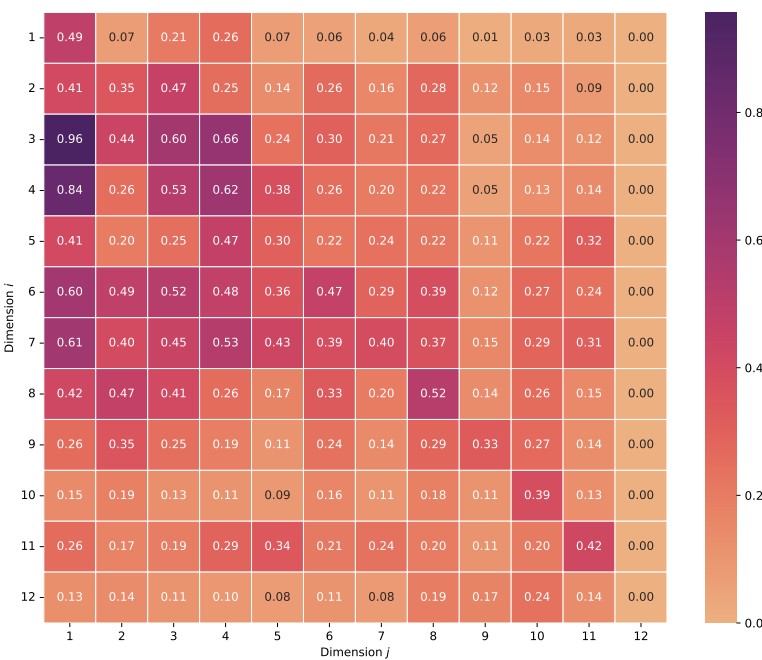

Figure 14: Predicted descendents matrix $\hat{D}^{\text{CF}}$ for RM in the 2024–2025 season.

$\hat{\alpha}^{\text{CF}}_{12,9} = 0.05$ but $\hat{\alpha}^{\text{RM}}_{12,9} = 0.10$. If we compare the estimated dynamic to what actually happened in 2024–2025, we can understand this failure state.

Comparing Figures 11 and 13, in particular the ninth lines and columns highlights that the model failed to predict how effective Tchouameni (#6) and Modrić (#7) would turn out to be at passing to Mbappé, indeed $\hat{\alpha}^{\text{CF}}_{9,6} = 0.01$ and $\hat{\alpha}^{\text{CF}}_{9,7} = 0.00$ while, in reality $\hat{\alpha}^{\text{RM}}_{9,6} = 0.06$ and $\hat{\alpha}^{\text{RM}}_{9,7} = 0.04$. Instead,

the model predicts that Mbappé will depend on Garcia (#2) and Valverde (#8) for passes, i.e. that he will be dependent directly on his rear as he was in PSG as can be seen by $\hat{\alpha}_{9,2}^{\text{PSG}} = 0.12$, $\hat{\alpha}_{9,8}^{\text{PSG}} = 0.12$, and the rest of the ninth line on Figure 9. This comparison suggests Mbappé and his teammates adapted quite significantly to each other, leading to him effectively playing quite differently.

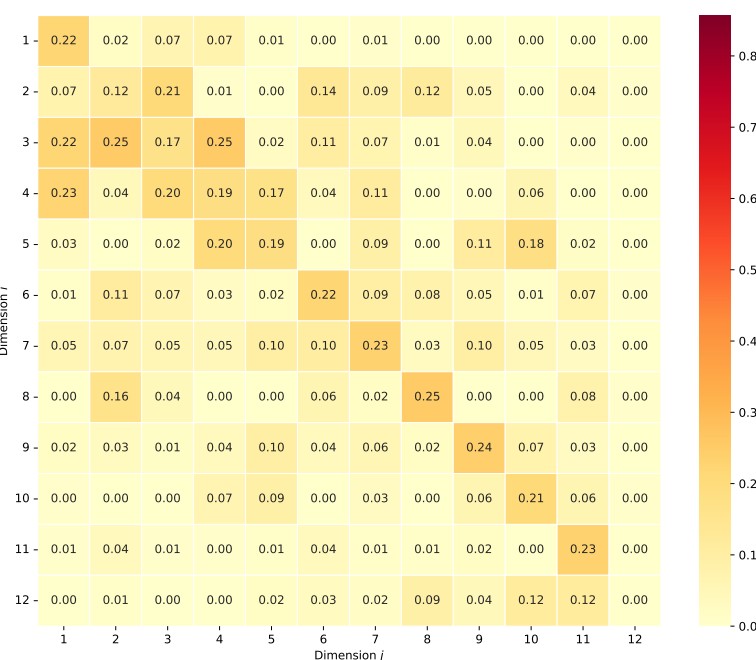

Figure 15: Predicted coefficients matrix $\hat{\alpha}^{\text{Liv}}$ for Liverpool in the 2024–2025 season.

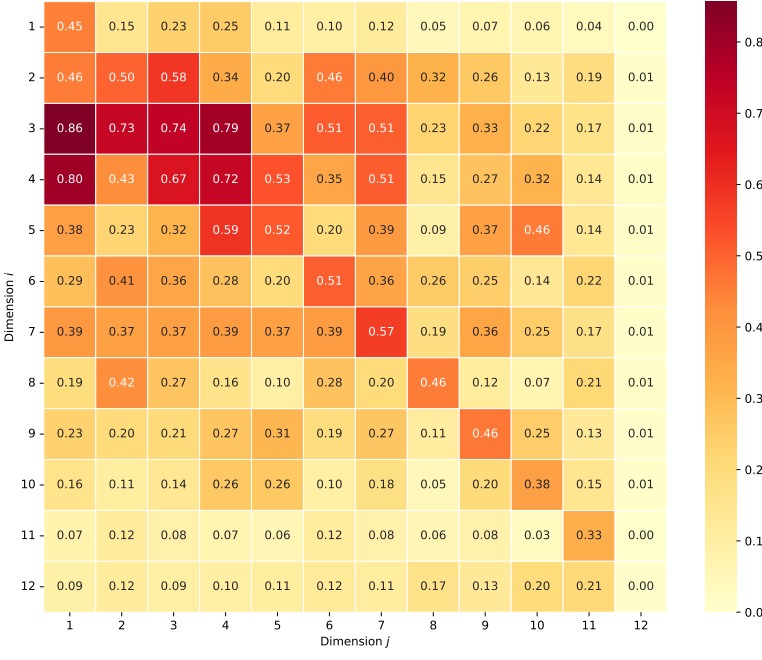

Figure 16: Predicted descendents matrix $\hat{D}^{\text{Liv}}$ for Liverpoool in the 2024–2025 season.

The similarity between this behaviour and the dynamics of the Liverpool team, and particularly Gakpo (#8), from Appendix D.5 suggests that Mbappé might be a good candidate to take up Gakpo's position. Performing the predictive attribution for this transfer in the same manner as previously, we obtain the coefficient and descendant matrices $\hat{\alpha}^{\texttt{Liv}}$ and $\hat{D}^{\texttt{Liv}}$ respectively. As noted, the model predicts Mbappé will have similar direct impact as in PSG, with $\hat{\alpha}^{\texttt{Liv}}_{12,8} = 0.09$ (just as $\hat{\alpha}^{\texttt{PSG}}_{12,8} = 0.09$), but this predicted performance is well below Gakpo's ($\hat{\alpha}_{12,8} = 0.09$ on Figure 6). Overall, the influence of Mbappé is estimated at $\hat{D}^{\texttt{Liv}}_{12,8} = 0.17$, well below Gakpo's $\hat{D}_{12,8} = 0.24$. While the model captures well the increased influence that Roberston (Liverpool's #2) could wield through Mbappé ($\hat{D}^{\texttt{Liv}}_{8,2} = 0.42$), it does not believe that Mbappé can take over Gakpo's dynamic in Liverpool, likely due to Mbappé's natural position being more of a central striker than an initiator like Gakpo. It is worth noting that, in general, certain potentially important variables—such as the off-field chemistry between players—are not taken into account by the model. The same applies to psychological factors affecting players, such as their adjustment to a new city or country. It is likely that access to such difficult-to-measure information would add value to the model.

