# OpenReview forum: "FeatHawkes: Scalable Feature-Based Attribution for Temporal Event Data"
_ICLR.cc/2026/Conference — ICLR 2026 Conference Withdrawn Submission_

### Official Review · Reviewer_hN9U · 2025-10-21

**Soundness:** 3
**Presentation:** 3
**Contribution:** 2
**Rating:** 6
**Confidence:** 3

**Summary:**

In their submission, the authors consider Hawkes Processes to model temporal count data. Their contributions are two-fold: 1) For the special (but widely used) case of an exponential kernel they propose a way to fit model parameters through stochastic gradient descent, which seems to show similar accuracy as state-of-the-art methods at a significantly improved runtime cost. 2) The consider the case where process parameters depend on external features and demonstrate (for the special case of a logistic regression) that these can be fit to data. In their experiments, they consider two synthetic datasets and one real-world dataset related to football analytics.

**Strengths:**

* The paper is clearly written and accessible.
* The authors clearly frame research gaps and own contributions.
* The authors clearly formalize the problem and provide a nice pedagogical introduction to the Hawkes process.
* Addressing the fitting procedure through SGD seems like an approach worthwhile trying. The authors nicely contextualize their proposed into the existing approaches for Hawkes processes.

**Weaknesses:**

* I am not convinced that the title of the paper actually captures the main contribution of the paper, which is the fitting procedure, rather than the attribution aspect. The attribution aspect receives little to no attention in the main text. Also, when introducing the Hawkes process, the authors put very little emphasis on the attribution aspect. There is a long and detailed discussion in the supplementary material, which goes into the direction of attribution, which is really interesting. But actually the main text should justify the title.
* The authors only mention Shapley values, but they could embed their attribution approach better into the rich literature on attribution methods in the XAI community e.g. [1],[2],[3], but only if they really want to emphasize the focus on attribution methods (see previous point).
* The authors mention causal attributions, but don't really pick up on this topic later. There is relevant literature in the XAI communinty e.g. [4],[5].
* The experimental part could be more comprehensive: As far as I understand, the authors used two single synthetic datasets (unclear if they really generalize to other parameter values) and a real-world football dataset, which does not seem to be publicly available. To illustrate the applicability of the approach, it would be helpful to include another real dataset ideally publicly available and from another application domain (the authors list many such application domains in the introduction).
* I find the experimental evaluation of the real-world example problematic. Unlike for the synthetic dataset, where there is a well-defined ground truth there is no such ground truth in this case. I don't think comparing parameter predictions for a simple and a more complex (feature-based) model is appropriate- in fact the simple model could be correct and the complex model could overfit to the data. It would be more appropriate to use other measures such as the approximation error etc to provide arguments why the more complex model is superior to the simple one.

The main weakness lies in the limited experimental validation and a lack of attribution insights.


[1] Scott M Lundberg and Su-In Lee. A unified approach to interpreting model predictions. In Advances in Neural Information Processing Systems, pages 4765–4774, 2017.
[2] Ian C Covert, Scott Lundberg, and Su-In Lee. Explaining by removing: A unified framework for model explanation. Journal of Machine Learning Research, 22(1):9477–9566, 2021.
[3] Wojciech Samek, Grégoire Montavon, Sebastian Lapuschkin, Christopher J Anders, and Klaus-Robert Müller. Explaining deep neural networks and beyond: A review of methods and applications. Proceedings of the IEEE, 109(3):247–278, 2021.
[4] Goyal, Y., Feder, A., Shalit, U., & Kim, B. (2019). Explaining classifiers with causal concept effect (cace). arXiv preprint arXiv:1907.07165.
[5] Alcaraz, J. M. L., & Strodthoff, N. (2024). CausalConceptTS: Causal attributions for time series classification using high fidelity diffusion models. arXiv preprint arXiv:2405.15871.

**Questions:**

* How severe is the restriction to Gaussian kernels? Are there commonly known examples where different parameterizations are used?
* At the end of 4.1, the authors discuss a direct comparison to Shapley values for the case where the true model follows a Hawkes process, which seems to provide an advantage for the Hawkes attribution as compared to the model-agnostic Shapley. It would actually be very interesting to see such a comparison for a real-world dataset that is not strictly a Hawkes process.

---

### Official Review · Reviewer_hn9A · 2025-10-22

**Soundness:** 1
**Presentation:** 2
**Contribution:** 1
**Rating:** 0
**Confidence:** 5

**Summary:**

This paper proposes an attribution method on top of Hawkes processes. The authors provide experiments with synthetic and soccer data to validate their proposed method.

**Strengths:**

1. Understanding how models work is an underexplored yet critical problems, especially for high-stake decision-making applications such as sports analytics.
2. This paper is easy to follow.
3. This paper provides additional analyses with soccer to interpret and discuss the attribution results.

**Weaknesses:**

1. While this paper presents interesting attribution applications to sports, it is an application-oriented paper that focuses on designing a method for the soccer setting and that develops an open-source library. I would recommend the authors consider submitting to data mining related conferences (e.g., KDD) given the limited nature and experiments mentioned below.
2. It is great that the authors present other similar domains that may involve similar scenarios in the first paragraph of the introduction; however, the proposed method is only evaluated with the soccer scenario, limiting the empirical generalizability to the corresponding claims.
3. It is insufficient with only discussing statistical-based attribution methods. This paper does not discuss and evaluate why existing attribution methods cannot tackle this task (note that the challenges in Sec. 2.4 are more related to Hawkes processes), e.g., [1, 2]. [1] is another attribution-based approach for badminton, which also tackles feature-level *predictive* attributions.
4. It remains unclear why the authors opt for the Hawkes process, where the second challenge in Sec. 2.4 is tailored to it. Based on the aforementioned concerns, this paper lacks technical novelty even though it will be open-sourced.
5. The synthetic experiments are too simplified and unclear about the motivation.
6. In L437-441, the authors explain the interpretation of having stable and unstable results. However, there is no further validation that can support the corresponding claims, e.g., how could the authors verify if the high errors are really because of underperformance during for those teams. Also, how would other temporal attribution methods perform? Similarly, in L1628, the authors mention *it does not believe that Mbappé can take over Gakpo’s dynamic in Liverpool, likely due to Mbappé’s natural position being more of a central striker than an initiator like Gakpo*. If the authors could not verify and clearly explain the reason, I have reservations that this tool could really convince coaches to trust it.

[1] ShuttleSHAP: A Turn-Based Feature Attribution Approach for Analyzing Forecasting Models in Badminton

[2] You Mostly Walk Alone: Analyzing Feature Attribution in Trajectory Prediction

[3] TimeSHAP: Explaining Recurrent Models through Sequence Perturbations

**Questions:**

Please see above

---

### Official Review · Reviewer_XMDa · 2025-10-31

**Soundness:** 3
**Presentation:** 2
**Contribution:** 2
**Rating:** 2
**Confidence:** 2

**Summary:**

The authors propose FeatHawkes, an algorithm for attributing outcomes to events in a point process. The authors propose a GPU-compatible implementation of this algorithm and benchmark it on synthetic data as well as a proprietary football dataset.

**Strengths:**

1. Level of technical description is very high; it is easy to follow the techique and evaluation protocol from the paper and appendix.
2. Paper is overall clearly written.
3. The software will be available under an open source license and the code is attached to the submission. The API looks clean and simple to use.

**Weaknesses:**

1. Discussion of prior work is very limited. Main comparison is vs. Ogata (1978) and Veen & Schoenberg (2008), vs. more recent work. The authors should justify this choice, but ideally expand to more recent literature.
2. The numerical studies are very superficial. 4.1. uses a single data generating process, same for 4.2 (which on top is a proprietary dataset). The authors should use public benchmark, and rigorously compare against prior work. Right now, the work is very anecdotal.
3. Terminology and interpretation is unclear, e.g. in Table 1. What do the different terms mean in the context of the attribution problem? This needs to be more clear from table/caption, or the referencing text.
4. Many figures and tables and their results (Figure 3, Table 1) are hard to place into context as only FeatHawkes was run. There are not methods from the literature, or some naive baseline to compare to judge the right magnitude of the numbers.

**Questions:**

1. In Figure 1, it seems like the estimation performance is very much on par with existing algorithms. Am I reading correctly that the speed gain in panel (b) is the main contribution of the work?
2. What would be a suitable baseline in Table 1? What is the correct way to interpret the results?

---

### Official Review · Reviewer_ypRi · 2025-11-01

**Soundness:** 3
**Presentation:** 3
**Contribution:** 2
**Rating:** 6
**Confidence:** 2

**Summary:**

This paper presents a method for credit attribution in time series that goes beyond last touch attribution by modeling the problem as a Hawkes process. The authors propose a specific formulation of the Hawkes process that can be computed via stochastic gradient descent, making it GPU-compatible and enabling better scalability compared to previous approaches. To validate their method, the authors benchmark it against synthetic use cases, demonstrating that it operates faster than existing approaches and scales more effectively. Finally, the authors apply their method to a real-world football dataset that they compiled.​​​​​​​​​​​​​​​​

**Strengths:**

* The authors propose a new formulation of the Hawkes process and provide an efficient algorithm to estimate it.
* On synthetic experiments with known ground truth, they demonstrate that their method can scale to more features and achieve smaller estimation errors.
* The authors apply their method to a real-world dataset and quantitatively show what analyses can be performed with it.
* On real-world data, they also compare their approach with a featureless model.
* In the appendix, they include a case study demonstrating the predictive power of credits assigned with their method.​​​​​​​​​​​​​​​​

**Weaknesses:**

* The real-world evaluation is conducted only for one sport—football—and while the authors describe that Hawkes process applications can be wide-ranging, it is not clear if the proposed formulation is suitable for other domains.
* The authors show through a case study what information and conclusions can be drawn from knowing the parameters of the Hawkes process. However, they do not provide any quantitative assessment.
* It is unclear how more accurate results translate to any downstream tasks, as knowing the parameters of the process might not be an end task in itself.
* Similarly, regarding the quantitative assessment of predictive power, an aggregation of some metric over multiple datasets would be useful.​​​​​​​​​​​​​​​​

**Questions:**

* Do you know of any real-world datasets where the ground truth is known or where there are clearly defined downstream tasks?
* Up to how many features does your method scale
* Shapley values are well understood in terms of their axioms. Does attribution from this process follow some of these axioms, such as null player or symmetry, or perhaps others?​​​​​​​​​​​​​​​​

---

### Note · Authors · 2026-04-24

I have read and agree with the venue's withdrawal policy on behalf of myself and my co-authors.

---

### Meta-Review · Area_Chair_JUWo · 2026-01-05

**Summary:**

This paper introduces FeatHawkes, a feature-augmented Hawkes process framework for event-level attribution in continuous time. Unlike traditional last-touch attribution, FeatHawkes models the problem using a Hawkes process and incorporates external features to enhance expressiveness. A key contribution is a novel first-order optimization method that uses stochastic gradient descent, enabling scalability with both large datasets and high-dimensional features, and making the framework compatible with modern machine learning pipelines.

**Reviewer Concerns:**

Reviewer ypRi has noted that real-world evaluation is conducted only on football data, that quantitative assessment of the attribution results was not provided, and that it is unclear how improved parameter estimation leads to downstream tasks. Reviewer XMDa explains that there is little discussion of prior work, the numerical studies are superficial and anecdotal, figures and tables are hard to interpret without baselines or comparisons to existing methods. Reviewer hn9A considers the work application-oriented and focused on a single soccer setting, finds the empirical evaluation insufficient to support broad claims, criticizes the lack of discussion and comparison with existing attribution methods, and questions the technical novelty and validity of the interpretations.

**Reviewer Scores:**

The authors did not provide a rebuttal to the comments made by the reviewers, so the reviewers may not change their score.

---

### Decision · Program_Chairs · 2026-01-26

Reject